# Roles of 670 nm Photobiomodulation on Rat Anterior Ischemic Optic Neuropathy: Enhancing RGC Survival, Mitochondrial Function, and Anti-Inflammatory Response

**DOI:** 10.3390/antiox14070886

**Published:** 2025-07-18

**Authors:** Tu-Wen Chen, Yao-Tseng Wen, Pei-Kang Liu, Monir Hossen, Rong-Kung Tsai

**Affiliations:** 1Institute of Eye Research, Hualien Tzu Chi Hospital, Buddhist Tzu Chi Medical Foundation, Hualien 97403, Taiwan; qooqoo700@tzuchi.com.tw (T.-W.C.); ytw193@tzuchi.com.tw (Y.-T.W.); monir.mbg@tzuchi.com.tw (M.H.); 2Department of Ophthalmology, Kaohsiung Medical University Hospital, Kaohsiung 80424, Taiwan; pkliu@kmu.edu.tw; 3School of Medicine, College of Medicine, Kaohsiung Medical University, Kaohsiung 80424, Taiwan; 4Institute of Biomedical Sciences, National Sun Yat-sen University, Kaohsiung 80424, Taiwan; 5Institute of Medical Sciences, Tzu Chi University, Hualien 97403, Taiwan; 6Doctoral Degree Program in Translational Medicine, Tzu Chi University and Academia Sinica, Hualien 97403, Taiwan

**Keywords:** 670 nm, anterior ischemic optic neuropathy, retinal ganglion cell survival, mitochondrial function, oxidative stress, anti-inflammatory

## Abstract

Non-arteritic anterior ischemic optic neuropathy (NAION) leads to retinal ganglion cell (RGC) loss and visual impairment, with no effective treatment. This study investigated the neuroprotective effect of 670 nm photobiomodulation (PBM) in a rat NAION model (rNAION). Wistar rats received 670 nm light exposure (10-min, 3000 lux) twice daily for 3 days after rAION injury, followed by 4 days of light treatment once a day. This study evaluated the neuroprotective effects of 670 nm light in an rNAION model. Rats received 670 nm light therapy (10 min/day, 3000 lux) for seven days post-injury. Treatment improved visual function (a 3.36-fold increase in FVEP amplitude), enhanced RGC survival (1.55-fold), and reduced apoptosis (a 15.86-fold reduction in TUNEL-positive cells). Inflammatory cytokines and ED1+ macrophage infiltration were significantly decreased. Oxidative stress was attenuated, with increased ATP, Nrf2, and PGC-1α levels and improved mitochondrial dynamics. These findings support 670 nm light as a potential therapy for NAION.

## 1. Introduction

The acute ischemic optic neuropathy known as non-arteritic anterior ischemic optic neuropathy (NAION) typically affects people over the age of 50 [1,2]. It is clinically characterized by painless vision loss, impaired color vision, and visual field defects associated with optic disc swelling, which eventually leads to optic disc atrophy [3]. The prevalence in Taiwan and the United States is estimated to be approximately 2.3–10.3 cases per million people. In Taiwan, the annual incidence is estimated to be 3.72 cases per 100,000 persons [4,5]. There are no effective treatments currently established for clinical NAION [1,2,3,4,5].

A disturbance of small vascular autoregulation in the posterior ciliary circulation is thought to be the cause of NAION, as it leads to inadequate vascular circulation around the optic nerve head (ONH) [4]. ON ischemia induces a sequence of harmful events, eventually resulting in retinal ganglion cell (RGC) loss [3]. RGC death and axon degeneration are the major consequences of ischemia brought on by oxidative stress [6,7,8], pro-inflammatory factors [9], and macrophage polarization [10]. Inflammation following an ON infarct includes macrophage infiltration and the disintegration of the blood–optic nerve barrier (BONB) [11]. This secondary degeneration results in the loss of RGCs. Therapeutic approaches for NAION aim to reduce neuroinflammation by lowering the expression of proinflammatory cytokines, modulating macrophage/microglia polarization, and inhibiting macrophage infiltration [9].

The ON and retina endure deleterious effects during ON ischemia including oxidative stress contributing to the degeneration of RGCs [12]. Excessive oxidative stress can lead to cellular damage, mitochondrial dysfunction, and impaired DNA repair systems, according to prior studies [13,14]. These are all well-known risk factors for the accelerated aging process and the development of neurodegenerative diseases [15,16]. As a result, one of the main therapeutic interventions is to reduce oxidative stress. The pathogenesis and features of rNAION are similar to those of human and primate AION, where laser-based photodynamic experiments produced superoxide radicals in the ON capillaries, leading to capillary thrombosis, inflammation, and RGC apoptosis [17].

Although several experimental studies have investigated potential therapeutic strategies for NAION, no clinically approved treatment currently exists. Oxidative stress, neuroinflammation, and mitochondrial dysfunction are key contributors to RGC degeneration following ischemia [12]. For example, the compound E212 improved visual outcomes by attenuating blood–retinal barrier disruption and neuroinflammation [18]. Similarly, G-CSF combined with meloxicam exhibited neuroprotective effects via Akt1 activation [19]. Furthermore, intravitreal G-CSF administration was shown to reduce M1 macrophage infiltration and protect microglia and RGCs after rNAION induction [9,20].Dietary vitamin B3 supplementation was shown to reduce oxidative stress and preserve vision in rNAION [21]. Astaxanthin also mitigated RGC apoptosis and preserved visual function [22]. Other agents, including soluble P-selectin [23], and omega-3 polyunsaturated fatty acids [10], have also demonstrated anti-inflammatory and anti-apoptotic properties in rNAION models.

Despite these promising preclinical findings, none of these approaches has translated into standardized, effective clinical therapies. Limitations include inconsistent efficacy, timing constraints, and a lack of large-scale validation.

PBM is a non-invasive therapeutic approach with well-documented benefits across various pathological conditions, primarily through mitochondrial signaling pathways that enhance ATP production and cellular survival [24]. Cytochrome c oxidase (CcO), a key enzyme in the mitochondrial electron transport chain, is a major intracellular photoacceptor for PBM and absorbs red to near-infrared light (600–1100 nm) [25,26]. Upon activation, it enhances the mitochondrial membrane potential, ATP production, and signaling molecules such as cAMP, thereby promoting cellular energy and function [27,28].

Notably, the absorption spectrum of oxidized CCO includes two peaks at 670 nm and 830 nm, whereas 728 nm shows minimal efficacy. Wong-Riley et al. [29] found that 670 nm light, when absorbed by primary visual cortex neurons, significantly increased metabolic activity and ATP production, supporting its role as an effective PBM wavelength. This mitochondrial activation underlies the neuroprotective and anti-inflammatory outcomes observed in prior studies.

A 670 nm PBM has demonstrated therapeutic effects in ocular diseases. Studies have reported improved outcomes in diabetic macular edema [30], vascular protection [31], and reduced retinal toxicity [32]. It also enhances mitochondrial function and reduces oxidative stress. Based on this evidence, we selected 670 nm to target RGC dysfunction in our rNAION model.

Therefore, the treatment of rNAION with 670 nm light may provide new therapeutic options. This study aimed to investigate the effects of 670 nm light therapy on the inflammatory response, oxidative stress, and apoptosis in the retinas of rats following AION induction.

## 2. Materials and Methods

### 2.1. Study Design

In the optimal therapeutic evaluation, 30 rats were evenly divided into 5 groups after the successful induction of AION (Figure 1A): control (sham group), AION induction without treatment (AION group), and 670 nm light treatment groups (RL groups), which received 10-min (RL-10), 20-min (RL-20), or 30-min (RL-30) of light exposure, respectively. The treatment was administered twice daily for 3 days after AION induction, followed by once-daily treatment for 4 additional days. To ensure uniform retinal illumination, 0.5% tropicamide and 0.5% phenylephrine were applied for mydriasis before 670 nm light exposure. The sham control group underwent the same procedures but was not exposed to 670 nm light. FluoroGold retrograde labeling was conducted on day 21 post AION to allow for sufficient tracer uptake and transport. FVEP recordings were subsequently performed on day 28, enabling the evaluation of visual pathway function in the same rats.

Light exposure at 670 nm for 10 min was chosen as the optimal treatment time in the rNAION model. Then, 45 rats were divided into 3 groups (Figure 1B): sham, AION, and RL-10 to further investigate the underlying mechanisms. The RL-10 group received 10 min of light exposure (twice a day for 3 days after AION injury, followed by 4 days of light treatment once a day). All rats were sacrificed on day 28 post-AION induction. Rat eyes were collected for ATP assay, IHC staining, TUNEL assay, qPCR, and Western blotting analysis. All of the rats tolerated this treatment and survived until the end of the procedure.

### 2.2. Animals

In this study, young male Wistar rats (4 weeks old) were used. The rats were purchased from BioLASCO Co. (Taipei, Taiwan) and housed individually in filter-top cages under standard laboratory conditions, with ad libitum access to food and water. The room temperature and relative humidity were maintained at 23 ± 1 °C and 55 ± 5%, respectively, under a 12-h light/dark cycle. All animal procedures were conducted in accordance with the ARVO Statement for the Use of Animals in Ophthalmic and Vision Research and were approved by the Institutional Animal Care and Use Committee (IACUC) of Tzu Chi Medical Center (Approval No. IACUC# 108-77). Prior to the surgical procedures, the rats were anesthetized via the intramuscular injection of ketamine (100 mg/kg) and xylazine (10 mg/kg) into the quadriceps muscle (both from Sigma, St. Louis, MO, USA).

### 2.3. AION Induction of Rat

The method for inducing the rNAION model was the same as that used in our previous reports [21,33]. Prior to general anesthesia, rats received topical Alcaine (Alcon, Puurs, Belgium) for anesthesia and Mydrin-P (Santen Pharmaceutical Co., Ishikawa, Japan) for pupil dilation. The rat was anesthetized via intramuscular injections of a mixture of ketamine and xylazine for general anesthesia. Subsequently, 2.5 mM Rose Bengal in PBS (1 mL/kg animal weight) was administered intravenously via the lateral tail vein of the rat. This route is widely accepted in laser-induced rNAION models that allows for rapid systemic distribution of the photosensitizer before laser activation of the ON head. After Rose Bengal injection, the optic disc was immediately exposed to an argon green laser system (MC-500 multi-color laser, Nidek Co. Ltd., Tokyo, Japan, setting: 532 nm wavelength, 500 µm size, and 80 mW power) for 12 pulses of 1 s each. A laser fundus lens (Ocular Instruments Inc., Bellevue, WA, USA) was used to focus the laser on the optic disc. Tobradex ointment was applied, and the rats were monitored until full recovery from anesthesia. The sham control group received a Rose Bengal injection without laser induction and was not subjected to 670 nm light exposure.

### 2.4. 670 nm Light Source Device and Treatment

Wistar rats received a topical instillation of 0.5% tropicamide and 0.5% phenylephrine hydrochloride (Santen, Osaka, Japan) eye drops. After eye drop administration, the rats were allowed to rest for 10–20 min to ensure full pupil dilation and were adapted to ambient light for 30 min prior to treatment [34]. Rats in the treatment group were generally anesthetized by the intramuscular injection of ketamine and xylazine to ensure complete immobilization as well as minimize stress and movement during the 670 nm treatment. The 670 nm light device used is shown in Figure 2. PBM was performed under strictly controlled environmental conditions to ensure reproducibility. During irradiation, anesthetized rats were placed in a cage (30 cm × 30 cm × 15 cm) containing wood shavings as bedding to minimize environmental stress. Two opposing wooden panels equipped with 670 nm LED array strips (DELTA ELECTRONICS INC, Taipei, Taiwan) were positioned to maintain a constant 2.5 cm distance between the irradiated eyes of the anesthetized rats and the light source. To ensure uniform illumination and minimize ambient light interference, a black, light-blocking cloth was used to cover the cage during irradiation. All procedures were conducted with the room lights turned off and without ambient lighting. This setup effectively minimized light scattering and ensured consistent light intensity throughout the entire exposure period (Figure 2).

The duration and frequency of the 670 nm light therapy was based on the well-characterized temporal dynamics of ischemia-induced cellular responses. Ischemic injury activates a cascade of signaling pathways, particularly those associated with oxidative stress, inflammation, and apoptosis, which are particularly elevated during the acute phase, typically within the first 72 h post-insult [17,35]. This window is considered critical for therapeutic intervention, as early and frequent modulation of these pathways may attenuate the secondary neuronal damage. Therefore, an intensified PBM regimen during this phase would enhance mitochondrial activation and antioxidant responses when tissue vulnerability is at its highest.

Specifically, we referenced Albarracin et al. [36] and Lu et al. [37], who applied 670 nm light at 50–60 mW/cm^2^ for 3 min per day over 5 consecutive days at a 2.5 cm distance, delivering 9 J/cm^2^ per session (total dose: 45 J/cm^2^), and demonstrated retinal protection in degeneration models. Similarly, Marco et al. [38] reported neuroprotective effects using 5 J/cm^2^ per session for 10 consecutive days (total dose: 50 J/cm^2^). For higher-dose comparisons, Begum et al. [39] administered 670 nm light at 20 mW/cm^2^ for 6 min, twice daily for 14 days (7.2 J/cm^2^ per session; total dose: 201.6 J/cm^2^) in an AMD model.

Building on previously reported dose–response studies, we established a stepwise PBM protocol to explore the optimal energy dose in our rNAION model. Accordingly, 670 nm light at 9 mW/cm^2^ was applied for 10, 20, or 30 min per session (equivalent to 5.4, 10.8, and 16.2 J/cm^2^, respectively), administered twice daily for 3 days and once daily for the remaining 4 days. The cumulative energy doses were 54 J/cm^2^ (RL-10 group), 108 J/cm^2^ (RL-20 group), and 162 J/cm^2^ (RL-30 group), respectively.

Rats were exposed to 670 nm light for 10, 20, and 30 min twice daily at 8:00 a.m. and 8:00 p.m. for 3 consecutive days. The light was delivered at an irradiance of 9 mW/cm^2^ at eye level, corresponding to energy fluences of 5.4, 10.8, and 16.2 J/cm^2^ per session, respectively. Subsequently, the same 670 nm light treatment was administered once daily at 8:00 a.m. for the following 4 days, delivering additional daily fluences of 2.7, 5.4, and 8.1 J/cm^2^, respectively. The sham group did not receive any 670 nm light treatment.

### 2.5. Retrograde Labeling of RGCs with FluoroGold and Morphometry of the RGCs

The detailed methods and protocol of FluoroGold (FG) retrograde labeling have been previously described [21,33]. On day 21 after rNAION induction, the rats were deeply anesthetized for retrograde labeling of the RGCs. The skin overlying the skull was cut open, and the lambda and bregma sutures served as landmarks for drilling holes. A total of 2 µL of 5% FG (Fluorochrome LLC, Denver, CO, USA) was injected into the superior colliculus on each side, at a depth of 4.5 mm from the surface of the skull, by a Hamilton syringe. The dye was injected at a point 5.5 mm caudal to the bregma and 1.5 mm lateral to the midline on both sides to allow for the retrograde transport of FG. Following injection, the burr holes were sealed with bone wax, and the skin incision was sutured. Rats were allowed to survive for 7 days post-labeling to ensure the sufficient retrograde transport of FG. Eyeballs were collected post-mortem for retinal preparation. After euthanasia, eyeballs were harvested and fixed in 10% formaldehyde for 1 h. Retinas were dissected, flattened by making four radial incisions, and mounted vitreous side upon microscope slides. RGCs were counted in six randomly selected fields from the central and midperipheral retina using a 400× magnification epifluorescence microscope (Axioskop; Carl Zeiss Meditech Inc., Thornwood, NY, USA) equipped with a digital imaging system. RGC densities were quantified using ImageMaster 2D Platinum software (version 7.0; GE Healthcare, Chicago, IL, USA).

### 2.6. Flash Visual Evoked Potentials (FVEPs)

The detailed protocol for FVEP recording has been described in our previous studies [21,33]. For electrode placement, the sagittal region of the skull was opened in the rats under anesthesia. Then, 4 mm screw implants were passed through the skull approximately 1.5 mm and placed at the frontal cortex and the primary visual cortex region of both hemispheres using stereotaxic coordinates. FVEPs were recorded using a commercial visual electrodiagnostic system (Diagnosys LLC, Lowell, MA, USA). The number of sweeps per average was 64 for each rat. A comparison of the amplitude of the P1–N2 wave in each group was made to evaluate visual function (*n* = 6 rats per group).

### 2.7. Sample Section Preparation

The detailed sample preparation procedure was conducted as previously described [21,33]. In brief, rats were euthanized after 28 days following rNAION induction. Eyes were enucleated along with the optic nerve (approximately 5 mm in length). The samples were fixed in 4% paraformaldehyde at 4 °C overnight. Subsequently, the eyeballs were dehydrated in 30% sucrose at 4 °C until they settled at the bottom of the tube. The eyeballs were then embedded in the OCT compound within plastic molds and rapidly frozen using liquid nitrogen. These frozen blocks were transferred to cryostats (−20 °C) and sectioned at a thickness of 16 µm for further analysis.

### 2.8. In Situ TdT-dUTP Nick End-Labeling (TUNEL) Assay

The TUNEL assay was performed according to the manufacturer’s instructions (Promega Corp, Madison, WI, USA), as previously described [21,33]. Briefly, three retinal sections per eyeball were rinsed in PBS to remove the OCT compound, and the tissues were incubated with proteinase K at room temperature for 10 min. Samples were then transferred to equilibration buffer and incubated with recombinant terminal deoxynucleotidyl transferase (rTdT) reaction buffer at 37 °C for 1 h. The reaction was terminated by immersing the slides in SSC solution for 15 min at room temperature, followed by 3 PBS washes. Nuclei were counterstained with 4, 6-diamidino-2-phenylindole (DAPI, 1:1000, Sigma-Aldrich, St. Louis, MO, USA), and coverslips were sealed. Images were captured from the central to mid-peripheral retina using a confocal microscope. TUNEL-positive cells in the RGC layer were manually counted in six randomly selected high-power fields (200×), and the average number per section was used for quantification.

### 2.9. Immunohistochemical Staining

ED1 is a marker for extrinsic macrophages, while 8-hydroxyguanine (8-OHdG) serves as an indicator of oxidative damage in both RNA and DNA, and TOMM20 is commonly used as a mitochondrial maker. Monoclonal antibodies of anti-ED1 (1:100, Catalog # ab283654, MCA341GA, Bio-Rad, Hercules, CA), -8-OHdG (1:200, Catalog # SAB5200010, Sigma-Aldrich, St. Louis, MO, USA), and TOMM20 (1:200; Catalog # MA5-32148, Invitrogen™, Eugene, OR, USA) were used in their procedure. The immunohistochemistry protocol was described in detail in our previous reports [21]. Briefly, cryosections were blocked with PBST containing 5% BSA at room temperature for 1 h. Primary antibodies were applied and incubated overnight at 4 °C. Sections were then incubated with secondary antibodies conjugated to fluorescein isothiocyanate (FITC) or rhodamine (1:100, Jackson ImmunoResearch Laboratories, West Grove, PA, USA) for 1 h at room temperature. Nuclear counterstaining was performed using 4, 6-diamidino-2-phenylindole (DAPI, 1:1000, Sigma-Aldrich, St. Louis, MO, USA). Fluorescence images were acquired using a microscope equipped with appropriate filter sets at ×200 magnification. Quantification of ED1-positive cells was performed using ImageMaster 2D Platinum Software V 7.0 (GE Healthcare, Chicago, IL, USA).

### 2.10. ATP Assay

The adenosine triphosphate (ATP) levels in rat retinal tissue were measured using an enzyme-linked immunosorbent assay (ELISA) kit for ATP (Cloud-Clone Corp, Houston, USA), following the manufacturer’s instructions. Briefly, retinal tissue was extracted from the eyeball, homogenized in fresh lysis buffer, sonicated, and centrifuged at 10,000× *g* for 5 min to obtain the supernatant. Briefly, retinal tissue was extracted from the eyeball, homogenized in fresh lysis buffer, sonicated, and centrifuged at 10,000× *g* for 5 min to obtain the supernatant. The ATP assay was conducted using standard, sample, control, and blank wells. A 50 µL aliquot of standard solution was added to the standard wells, 50 µL of retinal tissue supernatant to the sample wells, and 50 µL of standard dilution solution to the blank wells. Immediately after, 50 µL of Detection Reagent A was added to all wells, and the plate was incubated at 37 °C for 1 h. Following three washes, 100 µL of Detection Reagent B was added, and the plate was incubated for another 30 min at 37 °C. After five additional washes, 90 µL of substrate solution was added to each well and incubated at 37 °C for 10 min. The reaction was stopped by adding 50 µL of stop solution, and the absorbance was measured at 450 nm using a microplate reader. ATP concentrations were calculated based on a standard curve.

### 2.11. Retinal RNA Isolation and Real-Time PCR

Tissue RNA was extracted using the Qiagen RNeasy Mini Kit (Hilden, Nordrhein-Westfalen, Germany) from retinal or ON lysates obtained via sonication (130 W, 30% amplitude, 5 pulses for each sample, each pulse consisted of 5 s ON and 2 s OFF cycle). All RNA samples were reverse transcribed for 30 min at 42 °C with an iScript™ cDNA Synthesis Kit according to the manufacturer’s instructions (Bio-Rad, Hercules, CA, USA). Quantitative PCR (qPCR) was performed using Fast SYBR™ Green Master Mix (Thermo Fisher Scientific, Waltham, MA, USA) with specific primer pairs. Real-time RT-PCR was conducted using the ABI PRISM 7300 Sequence Detection System (Applied Biosystems, Waltham, MA, USA).

Using the NCBI Primer-BLAST tool, specific primers were designed to amplify fragments corresponding to the selected genes (Table 1). After triplicate amplification of each sample, the mean and standard error were calculated. The 2^−ΔΔCT^ method was used to determine the expression levels of all genes relative to GAPDH. Genes showing changes without statistical significance were not labeled. Statistical significance is indicated as follows: * *p* < 0.05 (significant), ** *p* < 0.01 (highly significant), *** *p* < 0.001 (very highly significant), and **** *p* < 0.0001 (extremely significant). “ns” indicates not significant.

### 2.12. Western Blotting Analysis

A protein assay was performed using a BCA protein assay kit. For immunoblotting, 20 µg of protein was separated on a 4–10% gradient gel (Invitrogen, Waltham, MA, USA) using 1X NuPAGE MOPS running buffer. Samples were loaded in triplicate along with 5 µL of BLUelf prestained ladder (GeneTex, Irvine, CA, USA). Proteins were transferred from the gel to a polyvinylidene difluoride (PVDF) membrane using Invitrogen’s iBlot2 dry transfer system and preassembled transfer stacks. Membranes were blocked with 5% non-fat milk in 1X TBST buffer (20 mM Tris-base (pH 7.6), 0.5 M NaCl, 0.5% Tween 20) for 1 h at room temperature. After rinsing with TBST, membranes were incubated overnight at 4 °C with primary antibodies against PGC1α (1:1000, Abcam, Cambridge, UK, ab191838), Nrf2 (1:1000, Thermo Fisher, Waltham, MA, USA, PA5-27882), NFκB (1:1000, Santa Cruz, Dallas, TX, USA, sc-8008), Opa1 (1:1000, Cell Signaling Technology, Danvers, MA, USA, 80471S), GAPDH (1:1000, Cell Signaling Technology, 2118S), IL1β (1:1000, Abcam, ab315084), Caspase 3 (1:1000, Cell Signaling Technology, 9662S), Bax (1:1000, Cell Signaling Technology, 2772S), and cytochrome c (1:1000, Abcam, ab133504), all diluted in 5% BSA in 1X TBST. After two washes with TBST, membranes were incubated with horseradish peroxidase (HRP)-conjugated secondary antibodies specific to the appropriate host species for 1 h at room temperature. Detection was performed using the iBright fl1000 imaging system. Membranes were incubated with ECL complex (Immobilon Western Chemiluminescent HRP substrate) for 3 min. Band intensities were quantified using iBright Analysis software 4.0.0 (Invitrogen, Carlsbad, CA, USA), and the protein expression levels were normalized to GAPDH.

### 2.13. Statistical Analysis

All statistical analyses were performed using GraphPad Prism version 8.0 (GraphPad Software, San Diego, CA, USA). The statistical analyses were presented as the means ± standard deviation (SD). For comparisons of differences among groups, the non-parametric Mann–Whitney U-test was employed. Results with *p*-values < 0.05 were statistically significant.

## 3. Results

### 3.1. Treatment with 670 nm Light for 10 Min Program Preserves Visual Function

FVEPs were recorded to evaluate the visual function of rats in the sham, AION, and RL 670 nm-treated groups (RL-10, -20, and -30) at 4 weeks after rAION induction (Figure 1A). The P1–N2 wave amplitude, which reflects the optic nerve function, was used as the primary readout. The mean P1–N2 amplitudes were 86.81 ± 27.7 µV (sham), 22.92 ± 17.8 µV (AION), 76.98 ± 25.71 µV (RL-10), 28.54 ± 16.76 µV (RL-20), and 28.66 ± 12.79 µV (RL-30), respectively (Figure 3B). The P1–N2 amplitude in the RL-10 group was significantly higher than that in the AION group (76.98 ± 25.71 µV vs. 22.92 ± 17.8 µV; **p* < 0.05; Figure 3B) and not significantly different from the sham group (76.98 ± 25.71 µV vs. 63.54 ± 7.254 µV; *p* > 0.05). In contrast, no significant improvement in P1–N2 amplitude was observed in the RL-20 and RL-30 groups compared with the AION group (28.54 ± 16.76 µV vs. 22.92 ± 17.8 µV and 28.66 ± 12.79 µV vs. 22.92 ± 17.8 µV; *p* > 0.05). These findings suggest that the 10 min program of 670 nm treatment effectively preserved visual function and optic nerve conduction following rNAION induction.

### 3.2. Treatment with 670 nm Light for 10 Min Program Promotes Survival of RGCs

Retrograde FG labeling was used to assess the RGC density as only RGCs with intact axons were tagged. To evaluate the protective effect of 670 nm PBM on RGC survival, the RGC densities in the central retina were quantified at 4 weeks after rNAION induction (Figure 4A). The average RGC densities in the sham, AION, and RL-10 groups were 1652.3 ± 210.3/mm^2^, 587.8 ± 187.3/mm^2^, and 911.6 ± 253.5/mm^2^, respectively (Figure 4B). Compared with the sham group, the AION group exhibited a significant 2.8-fold reduction in RGC density. Notably, the RL-10 group demonstrated a 1.6-fold increase in RGC density relative to the AION group (* *p* < 0.05, Figure 4B), indicating that the 10-min 670 nm light treatment effectively preserved RGCs after rNAION induction.

### 3.3. Treatment with 670 nm Light for 10 Min Program Inhibits RGCs Apoptosis and Reduces the Expressions of Apoptotic Markers in the Retina

To validate the neuroprotective effect of 670 nm light therapy, a TUNEL assay was performed on retinal cryosections to quantify apoptotic cells in the RGC layer across the sham, AION, and RL-10 groups. Apoptotic cells in the RGC layers of each group are shown in representative photomicrographs by TUNEL staining (Figure 5A). The average numbers of TUNEL-positive cells in the RGC layer per high-power field (HPF) was 0.3 ± 0.3 in the sham group, 11.1 ± 5.7 in the AION group, and 0.7 ± 0.3 in the RL-10 group, respectively (* *p* < 0.05, ** *p* < 0.01; Figure 5B). Compared with the sham group, the AION group exhibited a 37-fold increase in apoptotic RGCs, whereas the RL-10 group showed a 15-fold reduction in TUNEL-positive cells relative to the AION group (* *p* < 0.05, Figure 5B), indicating that 670 nm light therapy markedly attenuated RGC apoptosis after rNAION induction.

To further investigate the molecular mechanisms underlying the anti-apoptotic effects of 670 nm light therapy, qPCR analysis was performed to assess the mRNA expression of apoptotic regulators including caspase-8 (Casp-8), caspase-9 (Casp-9), and p53 in retinal tissue (gene primers were shown in Table 1). Results showed that the RNA levels of Casp-8, -9, and p53 were significantly reduced by 2.45-, 5.6-, and 2.87-fold in the RL-10 group compared with those in the AION group (* *p* < 0.05, Figure 5C).

Furthermore, Western blot analysis revealed that the protein expression levels of Bcl-2-associated X (Bax) and caspase 3 (Casp-3) were markedly decreased in the RL-10 group compared with those in the AION group (Figure 5D). The protein expression levels of Bax and Casp-3 were decreased by 3.19- and 13.11-fold, respectively, after treatment with 670 nm light versus AION (* *p* < 0.05, Figure 5E).

Collectively, these results demonstrate that the 10-min daily 670 nm light treatment significantly downregulates both the transcriptional and translational expression of key pro-apoptotic markers, supporting its protective, anti-apoptotic effect on retinal ganglion cells in the rNAION model.

### 3.4. Treatment with 670 nm Light for 10 Min Program Inhibits Macrophage Infiltration and Reduces Inflammation in the Optic Nerve and Retina

Furthermore, we performed ED1 staining to detect ED1 expression in the ON, a marker for phagocytic macrophages and microglia [40]. Anti-ED1 antibodies react with both extrinsic macrophages and intrinsic microglia [41]. Four weeks after rAION induction, ED1 staining of ON sections was performed, and the representative image is shown in Figure 4A. The number of ED1-positive macrophages was found to be 1.8 ± 0.5, 40.8 ± 10.7, and 3.6 ± 3.5 cells/HPF in the sham, AION, and RL-10 groups, respectively. These results demonstrated a 22-fold increase in macrophage infiltration in the AION group compared with the sham group. However, treatment with 670 nm light for 10 min significantly reduced the macrophage infiltration into the ON by 11.33-fold (* *p* < 0.05, Figure 6B) compared with the AION group following ON infarction.

Moreover, qPCR was conducted to detect the RNA expression of tumor necrosis factor-alpha (TNF-α), interleukin 6 (IL-6), interleukin 1 beta (IL-1β), and macrophage mannose receptor (CD206) in the ON (gene primers were shown in Table 1). Results showed that the RNA levels of TNF-α, IL-6, and IL-1β (pro-inflammatory markers) were reduced by 1.93-, 2.1-, and 1.62-fold, respectively, following 670 nm light treatment compared with AION (* *p* < 0.05, Figure 6C). In contrast, the RNA level of CD206 (anti-inflammatory marker) was significantly increased by 2.04-fold in the RL-10 group compared with the AION group (* *p* < 0.05, Figure 6C). Additionally, the protein expression levels of nuclear factor kappa-light-chain-enhancer of activated B cells (NFκB), and IL-1β in the retinas were evaluated (Figure 6D). The results demonstrated that 670 nm light therapy significantly decreased NFκB and IL-1β expression by 1.63-fold and 7.14-fold, respectively, compared with the AION group (* *p* < 0.05, Figure 6E).

These results indicate that the 10 min 670 nm light program effectively inhibits macrophage infiltration and exerts anti-inflammatory effects in both the ON and retina following rNAION.

### 3.5. Treatment with 670 nm Light for 10 Min Program Decreased Oxidative Damage and Increased ATP Production by Increasing Mitochondria in the Retina

To evaluate the extent of DNA/RNA oxidative damage following AION induction and treatment at 670 nm, retinal sections from the control and experimental samples were stained with antibodies that recognize 8OHdG and 8OHG. The most prominent immunostaining was localized to cells within the GCL (Figure 7A). Compared with the AION group, the RL-10 group exhibited substantially fewer 8OHdG/8OHG positive cells in the GCL, as shown in the immunostaining images.

To further investigate whether 670 nm light increased the mitochondrial content in the GCL of the experimental rNAION model, retinal sections were stained with a monoclonal antibody against TOMM20, a mitochondrial marker. Immunostaining revealed that the RL-10 group had more TOMM20-positive cells in GCL compared with the AION group (Figure 7B).

Furthermore, retinal ATP concentrations were measured by ELISA. The ATP levels in the retina of the sham group, AION, and RL-10 groups were approximately 419.5, 220, and 341.9 ng/mL, respectively. The reduction in ATP observed in the AION group was statistically alleviated by the 670 nm light treatment (* *p* < 0.05, Figure 7C). The ELISA results demonstrated a significantly higher ATP concentration in the RL-10 group compared with the AION group.

These findings suggest that the 10 min 670 nm light program may increase mitochondrial abundance, maintain higher ATP concentrations, and exert antioxidative effects in the retina following rNAION.

### 3.6. Treatment with 670 nm Light for 10 Min Program Reduced Inflammation and Promoted Antioxidative Pathways by Increasing Mitochondrial Metabolism in the Retina

The antioxidant activity of 670 nm light exposure was assessed by analyzing the protein expression of nuclear factor erythroid-2-related factor 2 (Nrf-2), a key regulator of cellular defense against oxidative stress. As shown in Figure 6, exposure to 670 nm light for 10 min significantly increased the Nrf-2 protein levels by 2.45-fold compared with the AION group (* *p* < 0.05, Figure 8B), suggesting enhanced antioxidative protection.

To further evaluate the mitochondria-mediated antioxidant potential, we analyzed the protein expression levels of key mitochondrial markers: peroxisome proliferator-activated receptor γ coactivator 1 alpha (PGC1-α), a marker of mitochondrial biogenesis; optic atrophy 1 (Opa-1), a marker of mitochondrial fusion; and cytochrome c (cyto-c), a marker of mitochondrial apoptosis (Figure 8). The results showed a significant increase in the PGC1-α levels by 2.07-fold in the RL-10 group compared with the AION group (* *p* < 0.05, Figure 8B), indicating enhanced mitochondrial biogenesis. Similarly, Opa-1 expression was significantly upregulated by 1.38-fold following 670 nm light treatment compared with the AION group (* *p* < 0.05, Figure 6). Additionally, cytochrome c expression was significantly reduced by 3-fold following 670 nm light treatment compared with AION, indicating decreased mitochondrial apoptosis (* *p* < 0.05, Figure 8B).

These findings suggest that 670 nm light exposure enhances antioxidant defense mechanisms, protects against oxidative damage, and promotes mitochondrial stability in the RGC layer following rNAION induction.

## 4. Discussion

### 4.1. Neuroprotection

NAION leads to ischemic axonopathy, resulting in RGC axon infarction and subsequent neuronal death [4,42]. The primary cause of irreversible visual impairment in ON injury is axonal degeneration and RGC loss, for which there is no effective treatment [43]. Given the crucial role of RGCs in visual processing and their high metabolic demand, promoting RGC survival is essential for therapeutic strategies [44,45,46]. Our study demonstrated that 670 nm PBM therapy significantly mitigated RGC loss compared with the AION group. Previous studies have reported that PBM reduces oxidative stress and inflammation while preserving retinal function in both human and animal models [47,48]. Therefore, 670 nm light improves retinal function and decreases RGC loss in a diabetic retinopathy model, aligning with our findings in rNAION [47].

### 4.2. RGC Survival

RGCs require substantial ATP to maintain action potential propagation due to their long axons and continuous visual signal transmission [49,50]. Mitochondrial dysfunction disrupts ROS homeostasis, leading to oxidative stress, cellular dysfunction, and apoptosis [51]. Our results indicate that the 10 min 670 nm light program of PBM therapy significantly enhanced the retinal ATP levels, increased the expression of the mitochondrial marker TOMM20, reduced oxidative stress, and decreased RGC apoptosis. Consistent with prior research, PBM facilitates mitochondrial cytochrome c oxidase (COX) activity, restoring ATP production and reducing ROS formation [52,53]. Clinical trials, such as NCT03866473, further explore the therapeutic potential of PBM in retinal disorders, reinforcing its translational significance [52].

### 4.3. Apoptosis Inhibition

Mitochondria-mediated apoptosis, primarily regulated by Bax and Cyto-c, plays a pivotal role in RGC degeneration following ischemic injury [54,55]. Bax activation induces mitochondrial membrane permeabilization, leading to Cyto-c release, apoptosome formation, and caspase cascade activation, ultimately resulting in cell death [56,57]. Opa-1 downregulation has been associated with ischemia-induced mitochondrial dysfunction and RGC loss [58]. Our results demonstrate that 670 nm light significantly reduces the expression of Bax, Casp-3, and Cyto-c while upregulating Opa-1, effectively inhibiting apoptosis and preserving mitochondrial integrity. These findings support the neuroprotective role of PBM in preventing mitochondrial dysfunction-associated apoptosis.

### 4.4. Anti-Inflammatory Effects

Neuroinflammation exacerbates NAION progression, with macrophage infiltration and microglial activation contributing to ON damage through the release of pro-inflammatory cytokines [9,21]. Our study revealed that 670 nm light significantly reduces macrophage infiltration (ED-1) and suppresses NFκB activation, leading to the decreased production of pro-inflammatory cytokines [59,60]. Additionally, 670 nm light treatment downregulates NFκB and IL-1β expression, consistent with reports indicating that NFκB inhibition reduces RGC loss in optic neuritis models [59].

### 4.5. Mitochondrial Function

Mitochondrial biogenesis and function are critical for neuronal survival and recovery following ischemic injury [61,62]. The Nrf-2/ARE signaling pathway regulates mitochondrial biogenesis by promoting mitochondrial DNA transcription through mitochondrial transcription factor A (TFAM), a downstream target gene of PGC1-α [63,64]. Previous research has demonstrated that the Nrf-2/ARE and PGC1-α pathways synergistically enhance neuroprotection and mitochondrial function [65]. Our results indicate that 670 nm light significantly upregulates Nrf-2, PGC1-α, and Opa-1 expression following rAION treatment, promoting mitochondrial biogenesis and fusion while preventing dysfunction-induced apoptosis. This finding aligns with previous studies linking Opa-1 downregulation to mitochondrial fragmentation and RGC loss [66,67].

### 4.6. Clinical Implications

PBM therapy offers a promising non-invasive intervention for rNAION by promoting RGC survival, reducing oxidative stress, suppressing neuroinflammation, and enhancing mitochondrial function. Our findings demonstrate that 670 nm PBM therapy improves retinal function by increasing RGC density, ATP levels, and mitochondrial integrity while decreasing apoptosis, inflammation, and oxidative damage. Additionally, 670 nm light upregulates Nrf-2 and Opa-1 while downregulating Bax and Cyto-c, reinforcing its neuroprotective role. Given PBM’s safety, cost-effectiveness, and ease of clinical translation [68], further studies should explore the optimal treatment parameters and potential combination therapies. Our observations imply that PBM mediates neuroprotection and may enhance RGC survival and ON recovery by improving mitochondrial function and reducing inflammation [34,69].

Ischemic injury is a common pathological mechanism underlying not only ocular diseases such as NAION, but also various central nervous system disorders including stroke, spinal cord injury, and neurodegenerative diseases such as Alzheimer’s disease (AD) and Parkinson’s disease (PD) [70,71,72,73].

Several studies have demonstrated that PBM with 670 nm light exerts neuroprotective effects beyond the visual system. For instance, Wong-Riley et al. reported that 670 nm PBM enhanced mitochondrial function and neuronal survival in primary cortical neurons following hypoxic injury [29]. Similarly, Detaboada et al. showed improved neurological recovery after traumatic brain injury in rodents using red/NIR light treatment [74]. Moreover, PBM has been shown to modulate microglial activation and reduce neuroinflammation in models of AD and PD [70].

These findings support the idea that the cellular and mitochondrial mechanisms triggered by PBM—such as enhanced ATP production, upregulation of antioxidant defenses, and modulation of inflammation—are conserved across multiple types of neurons and are not limited to RGCs. Therefore, while our current study focused on rNAION, the underlying mechanism of action suggests the broader potential applicability of 670 nm PBM in treating other ischemia-related neuronal injuries.

The lack of neuroprotection observed with the 20- or 30-min 670 nm PBM exposure in the rNAION model may be explained by the concept of “acquired resilience”, as proposed by Stone et al. [75]. This concept offers a compelling explanation for the observed dose-dependent effects of PBM. It describes an adaptive system in which low-dose physical or metabolic stressors (such as PBM) trigger endogenous protective responses that increase tissue resistance to subsequent injury. This system is mechanistically distinct from acquired immunity and involves non-specific, organism-wide responses including mitochondrial enhancement, the upregulation of antioxidant enzymes, mobilization of bone marrow-derived cells, and trophic cytokine release.

In the context of our study, 670 nm PBM at low energy densities may act as a mild metabolic stressor, activating these defense pathways and thereby preserving RGC function. However, as predicted by the principle of hormesis, this protective response is dose-dependent. Excessive exposure duration (20–30 min) may surpass the optimal stimulation window, leading to diminished efficacy or potential stress overload. This biphasic response may account for the lack of additional benefit—or even reduced efficacy—observed in the longer treatment groups.

Recent concepts of acquired resilience provide a plausible explanation for the biphasic dose–response observed in our PBM experiments. Low-dose stressors, such as short-duration 670 nm light exposure, may activate mitochondrial and anti-inflammatory pathways to protect retinal neurons, while prolonged exposure may surpass the optimal threshold and disrupt cellular homeostasis. These findings support the therapeutic principle of hormesis, which is widely recognized in resilience biology.

## 5. Conclusions

In conclusion, we demonstrated that a 10 min 670 nm light program provides neuroprotection in an rNAION model, both in morphometry and visual function. The actions are via anti-inflammation, anti-apoptosis, and antioxidation. The 670 nm light exposure ameliorated the damage caused by oxidative and ischemic stress after AION induction through its antioxidative and anti-apoptotic effects in the retina as well as anti-inflammatory effects in the optic nerve and retina (Figure 9). Our observations of 670 nm light exposure in rNAION may offer new non-invasive treatment options for patients with NAION.

## Figures and Tables

**Figure 1 antioxidants-14-00886-f001:**
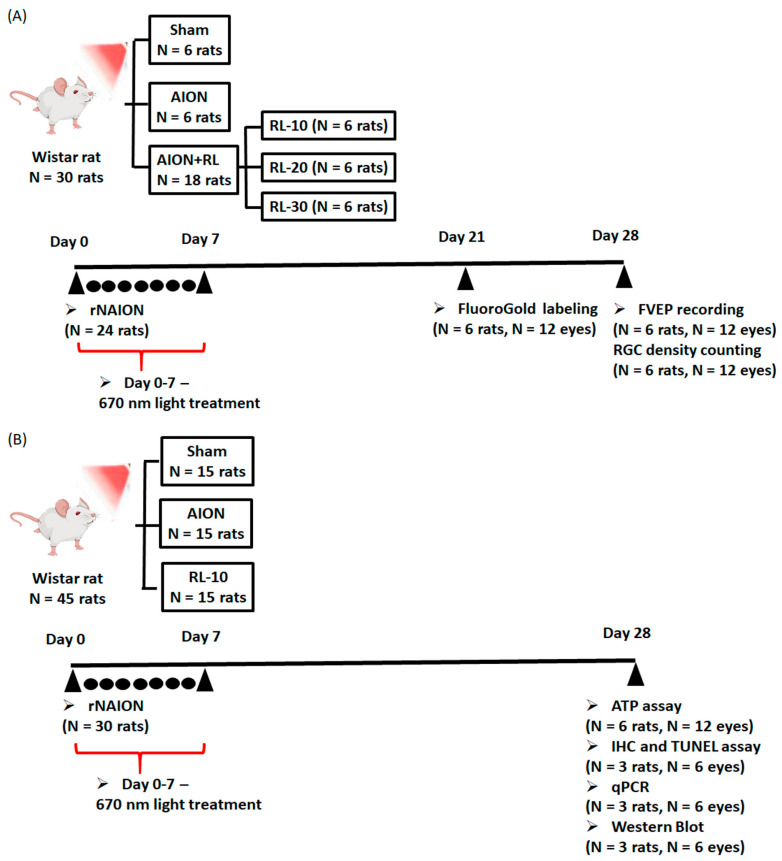
Schematic diagram illustrating the experimental workflow and PBM (670 nm) treatment schedule. (**A**) 30 rats were divided into sham, AION, and PBM groups (RL-10, RL-20, RL-30). PBM at 670 nm was applied twice daily for 3 days, then once daily for 4 days. FG labeling and FVEPs were performed on days 21 and 28, respectively. (**B**) Based on optimal results from (**A**), 45 rats were divided into sham, AION, and RL-10 groups to investigate underlying mechanisms. Retinal tissues were collected on day 28 for ATP assay, IHC, TUNEL, qPCR, and Western blot analyses.

**Figure 2 antioxidants-14-00886-f002:**
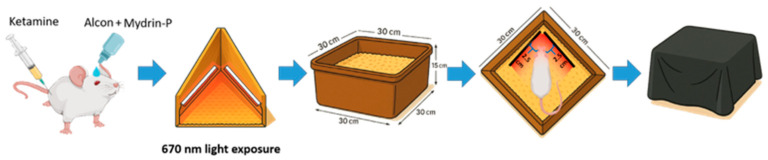
Schematic diagram of the 670 nm light source device for PBM treatment. The illustration depicts the positioning of the light source, the exposure distance (2.5 cm), and the treatment duration applied in the rNAION model.

**Figure 3 antioxidants-14-00886-f003:**
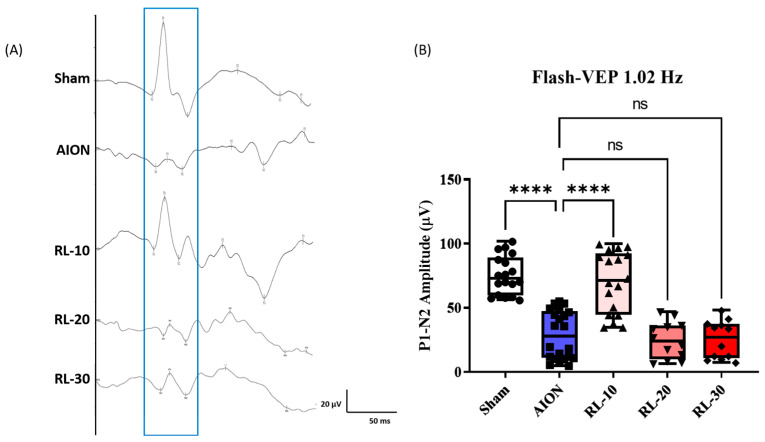
Protective effect of 670 nm red light on FVEPs at 4 weeks after rNAION induction. (**A**) Representative FVEP waveforms recorded at 4 weeks post-rNAION induction in the AION group and RL 670 nm-treated groups (RL-10, -20, and -30) compared with the control (sham) group. (**B**) Bar graphs showing the P1–N2 amplitudes of FVEPs in rats at 28 days post-rNAION. Amplitude values are presented as the mean ± SD (*n* = 12 eyes per group); **** *p* < 0.0001. ns, not significant. Scale bar = 20.0 µV and 50 ms.

**Figure 4 antioxidants-14-00886-f004:**
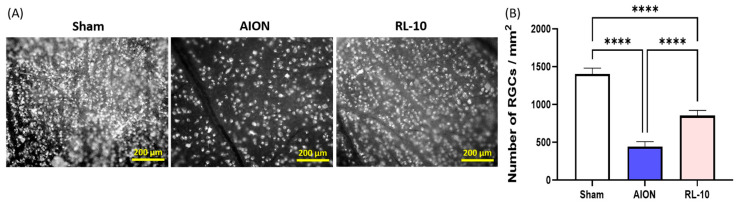
Protective effect of RL 670 nm on RGC survival in rNAION-induced rats at 28 days after rNAION induction. (**A**) Representative photomicrographs of flat-mounted central retinas showing RGC morphometry in each group via FG retrograde labeling at 21 days post-rNAION induction. FG-labeled RGCs were imaged at one-half retinal radius from the optic disk (white dots indicate surviving RGCs). (**B**) Quantification of FG-labeled RGCs in each group. Data are presented as the mean ± SD f (*n* = 12 eyes per group); **** *p* < 0.0001. ns, not significant. Scale bar = 200 µm.

**Figure 5 antioxidants-14-00886-f005:**
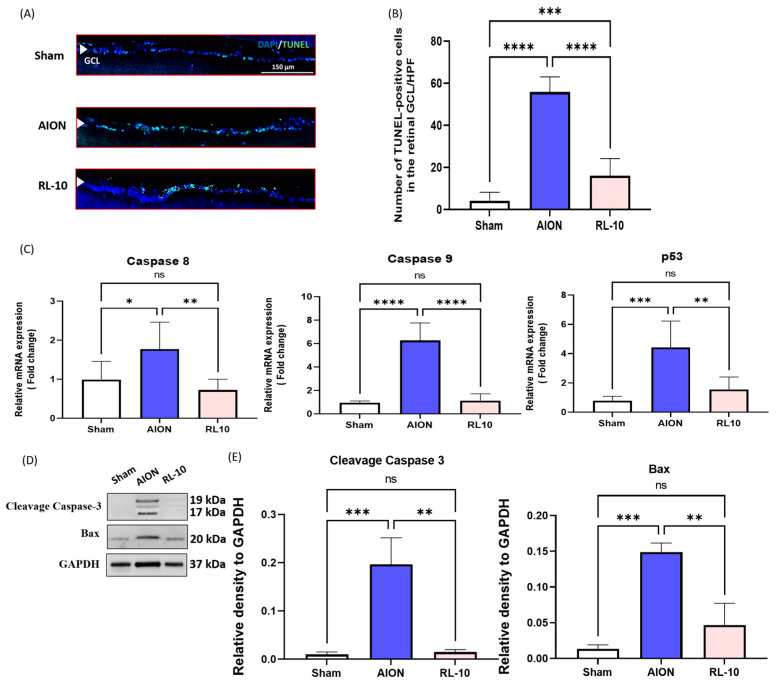
RL 670 nm repressed RGC apoptosis in the retinas of rAION rats at 28 days after rAION induction. (**A**) Representative photomicrographs showing apoptotic cells in the RGC layer of each group. TUNEL-positive cells (green) and nuclei counterstained with DAPI (blue) are shown in retinal sections at 28 days after rAION induction. (**B**) Quantification of TUNEL-positive cells per high-power field. Treatment with the 670 nm light for the 10 min program significantly reduced the number of apoptotic RGCs by 2.6-fold compared with the untreated group. (**C**) Relative mRNA expression levels of Casp-8, -9, and p53 in the retina at 28 days after rNAION induction. Each value was normalized to GAPDH. Expression levels of Casp-8, -9, and p53 were decreased in the RL-10 group compared with the AION group. (**D**) Representative Western blot image of retinal samples collected at day 28 post-rAION induction. (**E**) Quantification of Casp-3 and Bax protein bands. Protein expression levels were calculated using iBright imaging software 4.0.0 (Invitrogen, Carlsbad, CA, USA) and normalized to GAPDH. Casp-3 and Bax expression were significantly higher in the AION group (* *p* < 0.05; *n* = 6 eyes per group). Data are presented as the mean ± SD. ** *p* < 0.01, *** *p* < 0.001, **** *p* < 0.0001. ns, not significant. Scale bar = 150 µm.

**Figure 6 antioxidants-14-00886-f006:**
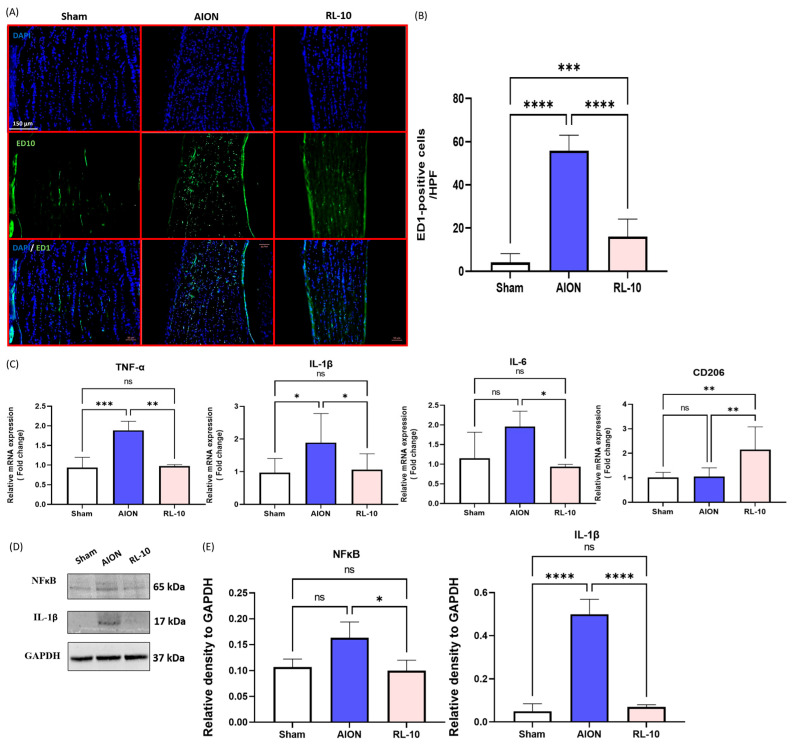
RL 670 nm repressed inflammatory macrophage infiltration in the ON of rNAION rats at 28 days after induction. (**A**) Representative photomicrographs of ED1-positive cells (green) and nuclei counterstained with DAPI (blue) in retinal sections at 28 days after rAION induction. (**B**) Quantification of ED1-positive cells per high-power field. (**C**) Relative mRNA expression levels of pro-inflammatory and anti-inflammatory markers in the ON at 28 days after rNAION induction. Each value was normalized to GAPDH. Expression levels of TNFα, IL-1β, and IL-6 were decreased, while CD206 expression was increased in the RL-10 group compared with the AION group. (**D**) Representative Western blotting images of retinal tissue collected at day 28 following rNAION induction. (**E**) Quantification of the NF-κB and IL-1β protein bands. Protein levels were calculated using iBright imaging software and normalized to GAPDH. NF-κB and IL-1β- expression levels were significantly higher in the AION group than in the sham group, but significantly reduced in the RL-10 compared with the AION group (* *p* < 0.05; *n* = 6 eyes per group). Data are presented as the mean ± SD. ** *p* < 0.01, *** *p* < 0.001, **** *p* < 0.0001.

**Figure 7 antioxidants-14-00886-f007:**
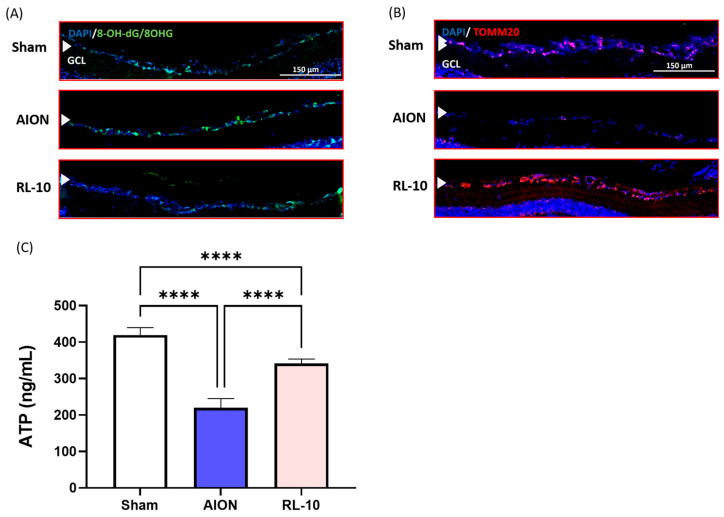
RL 670 nm repressed DNA/RNA oxidative damage while increasing the mitochondrial content and ATP levels of rAION rats at 28 days after induction. (**A**) Representative photomicrographs of 8-OXDG immunostaining (green) and nuclei counterstained with DAPI (blue) in retinal sections at 28 days after rAION induction. (**B**) Representative photomicrographs of TOMM20 staining (red), a mitochondrial marker, with DAPI-stained nuclei (blue) in retinal sections at 28 days after rNAION induction. (**C**) Retinal ATP levels in the sham, AION, and RL-10 groups analyzed by ELISA. A significant increase in ATP levels was observed in the RL-10 group compared with the AION group. Data are presented as the mean ± SD for each group (*n* = 12 eyes per group); **** *p* < 0.0001. Scale bar = 150 µm.

**Figure 8 antioxidants-14-00886-f008:**
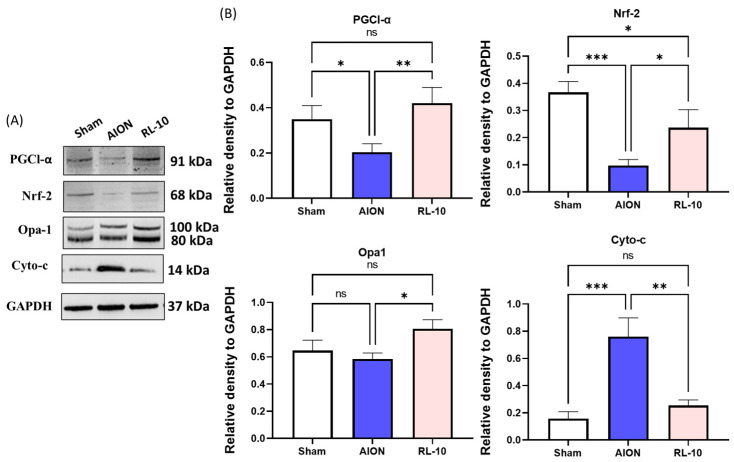
Analysis of retinal expression of PGC1-α, Nrf-2, Opa-1, and Cyto-c by immunoblotting. (**A**) Representative Western blot images of retinal tissue at day 28 following rAION induction. (**B**) Quantification of PGC1-α, Nrf-2, Opa-1, and Cyto-c protein bands. Each value was calculated using the iBright imaging program and normalized to GAPDH. The expressions levels of PGC1-α, Nrf-2, and Opa-1 were significantly higher in the RL-10 group compared with the sham and AION groups, while the expression of Cyto-c was significantly lower in the RL-10 group than in the sham or AION group (* *p* < 0.05, ** *p* < 0.01, *** *p* < 0.001, ns, not significant; *n* = 6 eyes per group). Data are presented as the mean ± SD.

**Figure 9 antioxidants-14-00886-f009:**
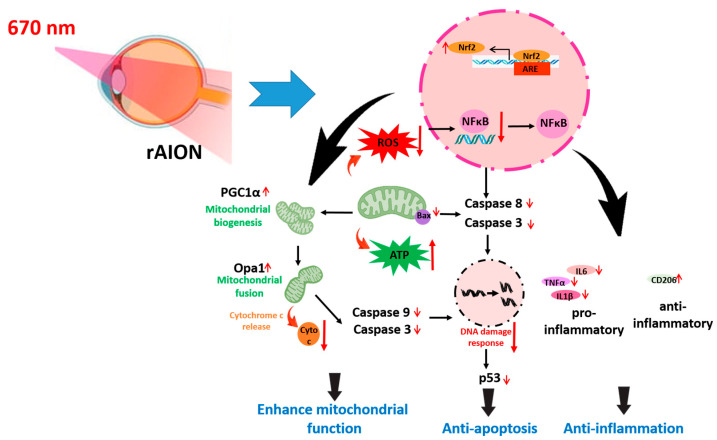
Summary of the neuroprotective effects of RL 670 nm in the rNAION model. A 10 min RL 670 nm treatment provided superior anti-inflammatory, antioxidative, and anti-apoptotic effects compared with the AION group.

**Table 1 antioxidants-14-00886-t001:** All of the gene primer information for qPCR.

Gene	Primer Sequences	Product Length (bp)	Annealing Temp (°C)	Gene Accession Number
Tnfa	ACCTTATCTACTCCCAGGTTCT/GGCTGACTTTCTCCTGGTATG	162	60	NM_012675.3
IL1B	TGCTGTCTGACCCATGTGAG/GTCGTTGCTTGTCTCTCCTTG	147	60	NM_031512.2
IL6	GCCCTTCAGGAACAGCTATGA/TGTCAACAACATCAGTCCCAAGA	153	60	NM_012589.2
CD206	TCOGTTTGCATTGCCCAGTA/AGAGTCTGTGCCCAAATCAAC	152	60	NM_001037168.1
Casp8	BGAGEEAGICECAAATCAAL/GCTGCTTCTCTCTTTGCTGAA	144	60	NM_022277.2
Casp9	AGCTGGCCCAGTGTGAATAC/GCTCCCACCTCAGTCAACTC	123	60	NM_001106130.1
p53	CACGAGCGCTGCTCAGATAGC/ACAGGCACAAACACGCACAAA	153	60	NM_030989.3
GAPDH	GATTTGGCCGTATCGGAC/GAAGACGCCAGTAGACTC	87	60	NM_017008.4

## Data Availability

The data presented in this study are available in the article.

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
