# Peer review of "Roles of 670 nm Photobiomodulation on Rat Anterior Ischemic Optic Neuropathy: Enhancing RGC Survival, Mitochondrial Function, and Anti-Inflammatory Response"

_antioxidants, 2025, doi:10.3390/antiox14070886_

Round 1

Reviewer 1 Report

I have thoroughly reviewed the manuscript by Chen et al. entitled “Roles of 670 nm Photobiomodulation on Rat Anterior Ischemic 2 Optic Neuropathy: Enhancing RGC Survival, Mitochondrial 3 Function, and Anti-Inflammatory Response”. The authors focus on a vision-threatening disease that, so far, lacks adequate treatment options. As such, the study holds significant relevance for both the scientific community and clinicians. While the study addresses an important issue and employs a variety of sophisticated methodologies, I do have some substantial concerns regarding the method descriptions, data presentation, and animal numbers.

2.2. Animals., line 89. Rats at the age of 3 weeks are not adult. Why have infant rats been used for this study? Maybe photobiomodulation exerts better effects in young than in adult rats? However, AION is a disease of adult/old subjects. Therefore , the use of adult/old rats would have been adequate.

2.3., line 98. Into which muscle was the mixture of ketamine and xylazine injected? Please specify.

2.3., line 100. Into which vein was rose bengal administered? Please specify.

2.4. Please specify the light treatment in more detail. Were the rats restrained? If yes, how, if the eyes were 2.5 cm away from the light source? In line 119 it should be rat eyes, not mouse eyes. Please specify the treatment of the sham group. Were these rats also treated with tropicamide and phenylephrine hydrochloride twice daily? Were the rats of the sham group also positioned close to a light source?

2.5. and 2.6. Were the same rats used for FVEP and for retrograde labeling?

2.13. Statistical analysis. If a nonparametric test (Mann–Whitney U-test) was used to compare the data, then the graphs should be presented as box plots.

Figure 3. An n=3 for Casp-3 and Bax- expression is too low to draw serious conclusions.

Figure 4. The representative pictures of the three groups (A) look very similar, but the graphs (B) differ extremely. The authors need to provide convincing pictures. An n=3 is too low to draw serious conclusions.

Figure 6. An n=3 is too low.

Author Response

Detailed comments

2.2. Animals., line 89. Rats at the age of 3 weeks are not adult. Why have infant rats been used for this study? Maybe photobiomodulation exerts better effects in young than in adult rats? However, AION is a disease of adult/old subjects. Therefore, the use of adult/old rats would have been adequate.

Ans:

    We appreciate the reviewer’s insightful comment. In our study, the rats were actually 4 weeks old. However, we utilized a well-established laser-induced rat anterior ischemic optic neuropathy (rNAION) model, which closely mimics the clinical and pathological features of human non-arteritic AION. rNAION has been established to express the most NAION features and symtoms after ischemic neuropathy (Bernstein, et al. 2011; Bernstein and Miller 2015; Cheng Zang, et al. 2009). This model involves photosensitizer (Rose Bengal) injection followed by laser activation at the ON head of rats, which results in localized vascular occlusion and secondary RGCs degeneration, distinct from age-related optic nerve changes. According to our previously studies (Hsu CL, et al. Int J Mol Sci. 2023 Jan 12;24(2):1504.; Chen TW, et al. Antioxidants (Basel). 2022 Dec 8;11(12):2422.) the mainstay of analysis experiments that can confirm the success if rNAION model include retrograde labeling of RGCs with FluoroGold and density of the RGCs, electrophysiological visual function by FVEP, apoptosis assays by TUNEL of the RGCs, and ED1 staining of the ONs.

    To specifically address the reviewer's concerns, we established a sham control group in which rats underwent the same procedure and injection of Rose Bengal but were not exposed to laser irradiation. These sham controls did not show RGCs loss or significant decreases in P1 or N2 wave amplitudes of the FVEP or inflammatory infiltration of the optic nerve, confirming that the pathological changes observed were not similar changes caused by age, but rather similar pathological changes caused by laser inducted, which is characteristic of the rNAION model.

2.3., line 98. Into which muscle was the mixture of ketamine and xylazine injected? Please specify.

Ans:

    We thank the reviewer for the important question regarding the anesthesia protocol. In our study, we used a commonly adopted anesthetic mixture of ketamine (100 mg/kg) and xylazine (10 mg/kg) for systemic anesthesia in rats, which provides sufficient depth and duration for ophthalmic procedures and laser-induced AION modeling. According to our previous standard protocol, this mixture was injected intramuscularly (IM) into the quadriceps muscle of rats as a means to achieve stable anesthesia during the experiment. We have described this in detail in Section 2.2. Animals.

2.3., line 100. Into which vein was rose bengal administered? Please specify.

Ans:

    We thank the reviewer for raising this important point. In our AION model, According to our previous standard protocol, rose bengal was administered via intravenous injection into the lateral tail vein of the rat. This is a standard and widely accepted route in laser-induced rNAION models that allows for rapid systemic distribution of the photosensitizer before laser activation of the ON head. We have described this in detail in Section 2.3. AION induction of rat.

2.4. Please specify the light treatment in more detail. Were the rats restrained? If yes, how, if the eyes were 2.5 cm away from the light source? In line 119 it should be rat eyes, not mouse eyes. Please specify the treatment of the sham group. Were these rats also treated with tropicamide and phenylephrine hydrochloride twice daily? Were the rats of the sham group also positioned close to a light source?

Ans:

    We thank the reviewers for their comments and for the opportunity to elucidate our phototherapy approach. We have added experimental workflow diagram shown in Figure 2, which presents a schematic diagram of the 670 nm light source setup used for PBM therapy. This figure also provides a detailed illustration of the experimental procedure, including the treatment duration, distance, and irradiation protocol applied in the rNAION model.

As shown in the Figure 1, the PBM was performed under carefully controlled conditions to ensure consistency and reproducibility. Specifically, during anesthesia, rats were placed in a cage with rodent bedding made of wood shavings to provide comfort and minimize stress. Two opposing wooden boards equipped with 670 nm light strips were placed at a fixed distance so that the irradiated eyes of the anesthetized rats were always kept at a precise 2.5 cm distance from the light source throughout the irradiation process. To ensure uniform illumination and prevent ambient light interference, a black, light-blocking cloth was used to cover the cage at the beginning of irradiation. This setup effectively minimized light scattering and maintained the stability of the light intensity during the entire exposure period. We have described this in detail in Section 2.4. 670 nm Light Source Device and Treatment

We thank the reviewer for pointing out this oversight. We have corrected the term from “mouse eyes” to “rat eyes” in line 170 of the revised manuscript to accurately reflect the animal model used in our study.

The sham group also treated with tropicamide and phenylephrine hydrochloride twice daily. The sham group was not exposed to the light source and was not placed near the irradiation device. These rats were kept in a separate area, away from the light device, to avoid any unexpected photobiological effects of the 670 nm light source.

2.5. and 2.6. Were the same rats used for FVEP and for retrograde labeling?

Ans:

We thank the reviewer for the insightful question. Yes, both FluoroGold retrograde labeling and FVEP were conducted on the same rats, in order to correlate RGCs density assessment and functional outcomes within the same subject. As for our experimental schedule, FluoroGold retrograde labeling was performed on day 21 after AION induction, allowing adequate time for tracer uptake and retrograde transport. FVEP recordings were subsequently performed on day 28, enabling evaluation of visual pathway function in the same rats. In accordance with our previous experimental protocols related to AION studies, the same rats were used for both retrograde labeling and FVEP recording. This allowed us to correlate functional outcomes with anatomical findings in the same individuals. We have added an experimental workflow diagram shown in Figure 1, which provides a clear schematic representation of the treatment timeline, assessment points, and the number of assays and biological samples used throughout the experimental procedure.

2.13. Statistical analysis. If a nonparametric test (Mann–Whitney U-test) was used to compare the data, then the graphs should be presented as box plots.

Ans:

We appreciate the reviewer’s valuable suggestion. In accordance with the recommendation, we have replaced the original bar graphs with box plots shown in Figure 3

Figure 3. An n=3 for Casp-3 and Bax- expression is too low to draw serious conclusions.

Ans:

Thank you to the reviewer for this important comment. We apologize for the confusion regarding sample size. Originally, the n values ​​in the article referred to the number of rats, and each Western blot analysis (including Casp-3 and Bax) was performed using retinal proteins extracted from 6 rat eyes per group (n = 6).

To improve the clarity of the article, we have now: provided a detailed experimental workflow diagram shown in Figure 1 to illustrate the experimental procedures, indicated the quantities used for each assay, and corrected the N values ​​in the figure legends and related text.

Figure 4. The representative pictures of the three groups (A) look very similar, but the graphs (B) differ extremely. The authors need to provide convincing pictures. An n=3 is too low to draw serious conclusions.

Ans:

  • We appreciate the reviewer’s constructive comment. In response, we have made

the following revisions: we have replaced the representative images in Figure 6A with more appropriate and convincing images that better reflect the quantitative differences shown in the statistical graphs shown in Figure 6B.

(B) Regarding the sample size: we would like to clarify that the n value in our study refers to the number of rat eyes, not animals. For IHC analysis, we collected six eyes per group (n = 6 rat eyes) for quantification. We regret the confusion caused by the previous description. To further improve transparency and help readers understand the overall workflow, we have included a supplementary schematic diagram outlining the experimental design and sample distribution, including the actual N values used for each assay shown in Figure 1.

Figure 6. An n=3 is too low.

Ans:

Thank you to the reviewer for this important comment. We apologize for the confusion regarding sample size. Originally, the n values ​​in the article referred to the number of rats, and each Western blot analysis (including PGC1-a, Nrf-2, Opa-1, and Cyto-c) was performed using retinal proteins extracted from 6 rat eyes per group (n = 6 rat eyes). To improve the clarity of the article, we have now: provided a detailed experimental workflow diagram shown in Figure 1 to illustrate the experimental procedures, indicated the quantities used for each assay, and corrected the N values ​​in the figure legends and related text.

The manuscript has been revised with attention to English language and readability. Figures were reviewed and adjusted to improve clarity and consistency where necessary.

Reviewer 2 Report

The paper "Roles of 670 nm Photobiomodulation on Rat Anterior Ischemic Optic Neuropathy: Enhancing RGC Survival, Mitochondrial Function, and Anti-Inflammatory Response" shows that 670 nm PBM can reduce neural degeneration in an animal model. Although this study does not introduce a new concept, I believe it is worth reporting as it confirms the effect in an animal pathology model. However, the manuscript focuses too heavily on the biological results, and lacks sufficient discussion on the physical aspects of the treatment. It is unclear why a specific method was chosen in terms of light wavelength range, duration, and treatment interval. As a result, there is no explanation provided for the 10-, 20-, and 30-minute treatment conditions. Additionally, since ischemia causes many neurological diseases, it would be beneficial to include a discussion on the potential scope of application—specifically, whether this treatment is effective only for ocular lesions or if it might also be applicable to other types of neuronal cells.

1. Introduction:
The authors mention that there is currently no effective treatment for NAION. For the readers’ understanding, it would be helpful to briefly introduce the existing treatment options that are being attempted and their limitations in the introduction.

2. Section 2.1 Study Design:
The animals should be clearly divided into five groups, and this information needs to be revised accordingly. It is also recommended to include a schematic diagram in the figure to illustrate the treatment and examination schedule.

3. Section 2.4 Treatment:
A diagram of the experimental setup would greatly enhance clarity. It is unclear whether the authors exposed the whole body of the animals to light.

4. Validation of rAION induction:
There is no experiment presented to confirm that rAION was successfully induced. A marker or indicator that verifies the induction of ischemia should be included. And the statements how the sham groups were treated should be inserted, since authors added several toxic chemical on the rat eye directly.. 

5.  Section 2.11 qPCR:
Reference or experiment is required to determine whether GAPDH is suitable as a reference gene.

6. FVEP data in Figure 1:
The FVEP data are critical for evaluating actual neural activity. However, the methods for peak detection and intensity measurement are not clearly explained. From the graphs in Figure 1, it appears that only a few peaks were chosen. It needs to be explained whether it is normal for large peaks to appear once or twice in this time interval.  Is it related to the animal movement? They were anethesized and other environments are same in this measurement? Further explanation is needed, possibly in supplementary materials.  Also, it is unclear how the sample size of n = 12 was determined. A more detailed description of how the animals were distributed across groups would be helpful.

7. Figure 2:
It should be confirmed whether the scale bars are consistent across all images. Each image in Figure 2 should include a scale bar for accurate interpretation.

8.  There are some typo in the manuscript..

Author Response

Major comments

The paper "Roles of 670 nm Photobiomodulation on Rat Anterior Ischemic Optic Neuropathy: Enhancing RGC Survival, Mitochondrial Function, and Anti-Inflammatory Response" shows that 670 nm PBM can reduce neural degeneration in an animal model. Although this study does not introduce a new concept, I believe it is worth reporting as it confirms the effect in an animal pathology model. However, the manuscript focuses too heavily on the biological results, and lacks sufficient discussion on the physical aspects of the treatment. It is unclear why a specific method was chosen in terms of light wavelength range, duration, and treatment interval. As a result, there is no explanation provided for the 10-, 20-, and 30-minute treatment conditions. Additionally, since ischemia causes many neurological diseases, it would be beneficial to include a discussion on the potential scope of application—specifically, whether this treatment is effective only for ocular lesions or if it might also be applicable to other types of neuronal cells.

Ans:

    We thank the reviewer for this thoughtful comment. The selection of the 670 nm wavelength for PBM in our study is grounded in both mechanistic and experimental evidence from previous peer-reviewed research. We have described this in detail in the Introduction, Section 2.4. 670 nm Light Source Device and Treatment, and Discussion.

We have described this in detail in Introduction

PBM is a non-invasive therapeutic approach with well-documented benefits across various pathological conditions, primarily through mitochondrial signaling pathways that enhance ATP production and cellular survival [26]. Cytochrome c oxi-dase (CcO), a key enzyme in the mitochondrial electron transport chain, is a major in-tracellular photoacceptor for PBM. It absorbs red to near-infrared light (600–1100 nm) [27,28], and upon activation, enhances mitochondrial membrane potential, ATP pro-duction, and signaling molecules like cAMP, thereby promoting cellular energy and function [29,30].

Notably, the absorption spectrum of oxidized CcO includes two peaks: one at 670 nm and another at 830 nm, while 728 nm demonstrates minimal efficacy. Wong-Riley et al. [31] found that 670 nm light, when absorbed by primary visual cortex neurons, significantly increases metabolic activity and ATP production, supporting its role as an effective PBM wavelength. This mitochondrial activation underlies the neuroprotec-tive and anti-inflammatory outcomes observed in prior studies.

670 nm PBM has shown therapeutic effects in ocular diseases. Studies have re-ported improved diabetic macular edema [32], vascular protection [33], and reduced retinal toxicity [34]. It also enhances mitochondrial function and reduces oxidative stress. Based on this evidence, we selected 670 nm to target RGCs dysfunction in our rNAION model.

    The PBM parameters used in our study—including treatment durations of 10, 20, and 30 minutes (administered twice daily for 3 days and once daily for the following 4 days)—were established based on the total energy dose (J/cm²) referenced from previously published protocols demonstrating the therapeutic efficacy of 670 nm light irradiation in models of retinal degeneration and age-related macular degeneration (AMD).

We have described this in detail in Section 2.4. 670 nm Light Source Device and Treatment

Specifically, we referenced Albarracin et al. [37] and Lu et al. [38], which applied 670 nm light at 50–60 mW/cm² for 3 minutes per day over 5 days at a 2.5 cm distance, delivering 9 J/cm² per session (total dose was 45 J/cm²), and demonstrated retinal pro-tection in degeneration models. Similarly, Marco et al. [39] reported neuroprotective effects with 5 J/cm² per session over 10 days (total dose was 50 J/cm²). For higher-dose comparisons, Begum et al. [40] administered 670 nm light at 20 mW/cm² for 6 minutes, twice daily for 14 days (7.2 J/cm² per session; total dose was 201.6 J/cm²) in an AMD model.

Building on these dose – esponse references, we established a graded PBM protocol to explore the optimal dose in our rNAION model. Accordingly, 670 nm light at 9 mW/cm² was delivered for 10, 20, or 30 minutes per session (equivalent to 5.4, 10.8, and 16.2 J/cm², respectively), administered twice daily for 3 days and once daily for the subsequent 4 days. The total energy doses were 54 J/cm², 108 J/cm², and 162 J/cm², re-spectively.

We have described this in detail in Discussion

Indeed, ischemic injury is a common pathological mechanism underlying not only ocular diseases such as NAION, but also various central nervous system disorders including stroke, spinal cord injury, and neurodegenerative diseases, such as Alzheimer’s disease (AD) and Parkinson’s disease (PD) [71-74].

Several studies have demonstrated that PBM with 670 nm light exerts neuropro-tective effects beyond the visual system. For instance, Wong-Riley et al. reported that 670 nm PBM enhanced mitochondrial function and neuronal survival in primary cor-tical neurons following hypoxic injury [31]. Similarly, Detaboada et al. showed im-proved neurological recovery after traumatic brain injury in rodents using red/NIR light treatment [75]. Moreover, PBM has been shown to modulate microglial activation and reduce neuroinflammation in models of AD and PD [71].

These findings support the idea that the cellular and mitochondrial mechanisms triggered by PBM—such as enhanced ATP production, upregulation of antioxidant de-fenses, and modulation of inflammation—are conserved across multiple types of neu-rons and not limited to RGCs. Therefore, while our current study focuses on rNAION, the underlying mechanism of action suggests broader potential applicability of 670 nm PBM in treating other ischemia-related neuronal injuries.

Detailed comments

  1. Introduction:

The authors mention that there is currently no effective treatment for NAION. For the readers’ understanding, it would be helpful to briefly introduce the existing treatment options that are being attempted and their limitations in the introduction.

Ans:

    We thank the reviewer for this valuable suggestion. In the revised Introduction section, we have added a brief overview of current therapeutic approaches for NAION. We have described this in detail in Introduction.

Although several experimental studies have investigated potential therapeutic strategies for NAION, no clinically approved treatment currently exists. Oxidative stress, neuroinflammation, and mitochondrial dysfunction are key contributors to RGC degeneration following ischemia [12]. For example, while the compound E212 im-proved visual outcomes by attenuating blood-retinal barrier disruption and neuroin-flammation [18]. Similarly, G-CSF combined with meloxicam exhibited neuroprotec-tive effects via Akt1 activation [19]. Furthermore, intravitreal G-CSF administration was shown to reduce M1 macrophage infiltration and protect microglia and RGCs af-ter rNAION induction [9,20].Dietary vitamin B3 supplementation was shown to re-duce oxidative stress and preserve vision in rNAION [21], astaxanthin mitigated RGCs apoptosis and preserved visual function [22]. Other agents, including algae oil [23], puerarin [24], soluble P-selectin [25], and omega-3 polyunsaturated fatty acids [10], have also demonstrated anti-inflammatory and anti-apoptotic properties in rNAION models.

Despite these promising preclinical findings, none of these approaches have translated into standardized, effective clinical therapies. Limitations include incon-sistent efficacy, timing constraints, and a lack of large-scale validation.

  1. Section 2.1 Study Design:

The animals should be clearly divided into five groups, and this information needs to be revised accordingly. It is also recommended to include a schematic diagram in the figure to illustrate the treatment and examination schedule.

Ans:

We thank the reviewer for this helpful suggestion. In the revised manuscript, we have clearly updated the description of the five experimental groups in Section 2.1. Study Design to improve clarity and reproducibility.

In addition, we have added a schematic diagram shown in Figure 1 that visually illustrates the group allocation, treatment timeline, and key examination time points, including irradiation periods and follow-up assessments. This figure provides a comprehensive overview of the experimental workflow and enhances the overall clarity of the study design.

  1. Section 2.4 Treatment:

A diagram of the experimental setup would greatly enhance clarity. It is unclear whether the authors exposed the whole body of the animals to light.

Ans:

We thank the reviewer for the helpful suggestion. In response, we have added a schematic diagram of the experimental setup shown in Figure 2 to clearly illustrate the arrangement of the animal during irradiation. We have described this in detail in Section 2.4. 670 nm Light Source Device and Treatment.

The rats in the treatment group were generally anesthetized by intramuscular injection of ketamine (100 mg/kg) and xylazine (10 mg/kg) to ensure complete immobilization and minimize stress and movement during 670 nm treatment. 670 nm light device was shown in the Figure 2, the PBM was performed under carefully controlled conditions to ensure consistency and reproducibility. Specifically, during anesthesia, rats were placed in a cage (30 cm × 30 cm × 15 cm) with rodent bedding made of wood shavings to provide comfort and minimize stress. Two opposing wooden boards equipped with a 670 nm LED array strips (DELTA ELECTRONICS INC, Taipei, Taiwan) were placed at a fixed distance so that the irradiated eyes of the anesthetized rats were always kept at a precise 2.5 cm distance from the light source throughout the irradiation process. To ensure uniform illumination and minimize ambient light interference, a black, light-blocking cloth was used to cover the cage during irradiation. The procedure was conducted with the room lights turned off and in the absence of any direct ambient lighting. This setup effectively minimized light scattering and maintained the stability of the light intensity during the entire exposure period.

  1. Validation of rAION induction:

There is no experiment presented to confirm that rAION was successfully induced. A marker or indicator that verifies the induction of ischemia should be included. And the statements how the sham groups were treated should be inserted, since authors added several toxic chemical on the rat eye directly.

Ans:

    We thank the reviewer for this important observation. In our study, rNAION was induced using a standardized and widely accepted method involving lateral tail vein injection of the photosensitizer Rose Bengal followed by laser induction at the optic nerve head, which results in localized vascular occlusion and secondary RGCs degeneration—hallmarks of AION pathology.

According to our previously studies (Hsu CL, et al. Int J Mol Sci. 2023 Jan 12;24(2):1504.; Chen TW, et al. Antioxidants (Basel). 2022 Dec 8;11(12):2422.) the mainstay of analysis experiments that can confirm the success if rNAION model include retrograde labeling of RGCs with FluoroGold and density of the RGCs, electrophysiological visual function by FVEP, apoptosis assays by TUNEL of the RGCs, and ED1 staining of the ONs. To confirm the successful induction of rNAION, we followed our previously published protocol and included multiple validated indicators:

  1. Retinal ganglion cell (RGC) loss, as counted by RGC density.
  2. Significant reductions in P1-N2 amplitudes of flash visual evoked potentials (FVEP).
  3. Increased RGCs apoptosis by TUNEL assay.
  4. Histological evidence of inflammatory infiltration at the optic nerve head.

These findings are consistent with the ischemic damage seen in established rNAION models and support the conclusion that rNAION was successfully induced in our experimental animals. We established a sham control group in which rats underwent the same procedure and injection of Rose Bengal but were not exposed to laser irradiation. These sham controls did not show retinal ganglion cell loss or significant decreases in P1 or N2 wave amplitudes of the visual FVEP or inflammatory infiltration of the optic nerve.

We have described this in detail in Section 2.1. Study Design

To ensure uniform retinal illumination, 0.5% tropicamide and 0.5% phenylephrine were applied for mydriasis before 670 nm light exposure. The sham control group re-ceived the same treatment without 670 nm light treatment.

  1. Section 2.11 qPCR:

Reference or experiment is required to determine whether GAPDH is suitable as a reference gene.

Ans:

We thank the reviewer for this comment. Here, GAPDH was chosen as the reference gene for qPCR normalization because, in addition to other studies conducted by Nitza Goldenberg-Cohen et al. (Investn Ophal 2005 Ophalvesta Mar;46(8):2716–2725), and in our previously study by Huang TL, et al. (Invest Ophthalmol Vis Sci. 2017 Mar 1;58(3):1628-1636.), GAPDH was reliably used as a housekeeping gene in qPCR assays in the rNAION model. This supports the rationale for using GAPDH as an internal control in our current experimental setting.

  1. FVEP data in Figure 1:

The FVEP data are critical for evaluating actual neural activity. However, the methods for peak detection and intensity measurement are not clearly explained. From the graphs in Figure 1, it appears that only a few peaks were chosen. It needs to be explained whether it is normal for large peaks to appear once or twice in this time interval.  Is it related to the animal movement? They were anethesized and other environments are same in this measurement? Further explanation is needed, possibly in supplementary materials.  Also, it is unclear how the sample size of n = 12 was determined. A more detailed description of how the animals were distributed across groups would be helpful.

Ans:

We thank the reviewer for this important comment. We will clearly describe the criteria used for FVEP peak detection and amplitude measurement as follows. The FVEPs were recorded at 28 days after rNAION induction and followed our previous reports (Chien, et al. 2016). FVEP records the electrical information from ganglion cell layer of retina to optic nerve and finally to visual cortex during diffuse flash stimulus (Weinstein, et al. 1991). The integrity of the entire visual pathway can be tested by FVEP system. Medically, the normal waveform is the first negative peak (N1), followed by a large positive peak (P1), and then a negative peak (N2). The amplitude of P1-N2 represents the number of functional neurons of retina and the prolongation of P1 latency represents the optic nerve function (Helga Kolb, et al. 1995). FVEP can be used to diagnose the demyelination and axonal pathologies in optic nerve. Moreover, FVEP can signal the demyelination phenomena which shows the changes in latency represents and alterations the amplitude in P1-N2 waveform (Baiano and Zeppieri 2022; Park, et al. 2022).

The P1–N2 amplitude in FVEPs reflects the number and functional status of retinal neurons, particularly RGCs. A reduction in P1–N2 amplitude indicates a loss of RGC function or number, as well as impaired ON conduction.

In Figure 1, for the AION group, the FVEP traces showed only a few small or diminished peaks, which is consistent with RGCs dysfunction or loss caused by ischemic optic neuropathy. The resulting decrease in P1–N2 amplitude reflects impaired visual signal transmission, confirming the successful induction of the rNAION model. This waveform pattern is typical and has been reported in our previous studies of AION-induced RGCs damage (Lin WN, et al. Mar Drugs. 2020 Jan 28;18(2):85.; Huang TL, et al. Mar Drugs. 2020 Jan 27;18(2):83.). In the RL-20 and RL-30 treatment groups, the P1–N2 amplitudes remained low and indistinct, suggesting that longer durations of 670 nm PBM (20 or 30 minutes) did not result in significant neuroprotective effects in rNAION model. As for the P1–N2 amplitude of FVEP in the RL-10 group was significantly higher than that in the untreated AION group and was similar to that in the Sham group.

We thank the reviewer for pointing out this oversight. In line 234, n = 12 rats per group is a typo. It should be n = 6 rats per group. We have corrected it in line 234 of the revised manuscript.

  1. Figure 2:

It should be confirmed whether the scale bars are consistent across all images. Each image in Figure 2 should include a scale bar for accurate interpretation.

Ans:

We thank the reviewer for this careful observation. We have carefully checked all images shown in Figure 4 and confirmed that the scale bars are consistent across all panels. In addition, we have added scale bars to each image to ensure that each image contains a clearly labeled scale bar for accurate interpretation and comparison. This issue has now been corrected in the revised version of Figure 4.

  1. There are some typo in the manuscript.

Ans:

We thank the reviewer for pointing this out. We have carefully proofread the entire manuscript and corrected all typographical and grammatical errors to improve clarity and readability.

The manuscript has been revised with attention to English language and readability. Figures were reviewed and adjusted to improve clarity and consistency where necessary.

Reviewer 3 Report

see above at he specific points

see above at the specific points, minor: 

  1. Why is the abbreviation of the model rAION when the abbreviation of the disease is NAION? (At some point it is written as rat NAION model, I think it would be more transparent if it was abbreviated as rNAION.)
  2. There are some typos: e.g. line 260: in the title function

Author Response

Is the research design appropriate and are the methods adequately described?

  1. For comparing three groups, the Mann-Whitney U-test is not suitable, e.g. Kruskal Wallis ANOVA is needed instead, but this was used everywhere.

Ans:

    We thank the reviewers for their statistical comments. This study used a test based on our previously established analytical method (Chen TW et al., Antioxidants (Basel 2022 Dec 8;11(12):24-22; Wen YT et al., Int J Mushroom Med 2022;24(2):41-48). This analytical method performed pairwise comparisons between groups (three groups were compared, including sham, rNAION, and treatment groups), and the number of groups was consistent with that described in the previous publication.

  1. When were the retinas isolated for Western blotting? 4 weeks later? Why was this time chosen? Literature suggests that signaling pathways activated by damage are more active earlier.

Ans:

    We agree that early activation of signaling pathways (e.g., oxidative stress, inflammation, apoptosis) is often more robust in the acute phase following ischemic injury. However, our aim was to assess whether PBM exerts a sustained regulatory effect on these pathways beyond the acute phase, reflecting its potential for long-term neuroprotection.

  1. Why did you use 3 week old animals (age for a child or a young adult human) when the condition you have described occurs over 50 years of age?

Ans:

We appreciate the reviewer’s insightful comment. In our study, the rats were actually 4 weeks old. However, we utilized a well-established laser-induced rNAION model, which closely mimics the clinical and pathological features of human NAION. This model involves photosensitizer (Rose Bengal) injection followed by laser activation at the ON head, which results in localized vascular occlusion and secondary RGCs degeneration, distinct from age-related optic nerve changes.

    To specifically address the reviewer's concerns, we established a sham control group in which rats underwent the same procedure and injection of rose bengal but were not exposed to laser irradiation. These sham controls did not show retinal ganglion cell loss or significant decreases in P1 or N2 wave amplitudes of the FVEP or inflammatory infiltration of the optic nerve, confirming that the pathological changes observed were not similar changes caused by age, but rather similar pathological changes caused by laser inducted, which is characteristic of the rNAION model.

  1. Why did the twice-daily treatment last for 3 days and then the once-daily treatment for 4 days? How were these time intervals chosen?

Ans:

    We thank the reviewer for this important question. The rationale behind the PBM treatment schedule—twice-daily administration for the first 3 days followed by once-daily treatment for the next 4 days—has been described in detail in Section 2.4. 670 nm Light Source Device and Treatment.

Duration and frequency of 670 nm light therapy was based on the well-characterized temporal dynamics of ischemia-induced cellular responses. Is-chemic injury activates signaling pathways related to oxidative stress, inflammation, and apoptosis, which are particularly elevated during the acute phase, typically within the first 72 hours post-insult [17,37]. This time window is regarded as critical for ther-apeutic intervention, as early and frequent modulation of these pathways may attenu-ate the cascade of secondary neuronal damage. Therefore, an intensified PBM regimen during this phase was designed to enhance mitochondrial activation and antioxidant responses when tissue vulnerability is highest.

Specifically, we referenced Albarracin et al. [38] and Lu et al. [39], which applied 670 nm light at 50–60 mW/cm² for 3 minutes per day over 5 days at a 2.5 cm distance, delivering 9 J/cm² per session (total dose was 45 J/cm²), and demonstrated retinal pro-tection in degeneration models. Similarly, Marco et al. [40] reported neuroprotective effects with 5 J/cm² per session over 10 days (total dose was 50 J/cm²). For higher-dose comparisons, Begum et al. [41] administered 670 nm light at 20 mW/cm² for 6 minutes, twice daily for 14 days (7.2 J/cm² per session; total dose was 201.6 J/cm²) in an AMD model.

Building on these dose – response references, we established a graded PBM proto-col to explore the optimal dose in our rNAION model. Accordingly, 670 nm light at 9 mW/cm² was delivered for 10, 20, or 30 minutes per session (equivalent to 5.4, 10.8, and 16.2 J/cm², respectively), administered twice daily for 3 days and once daily for the subsequent 4 days. The total energy doses were 54 J/cm², 108 J/cm², and 162 J/cm², respectively.

Rats were exposed to 670 nm light for 10, 20, and 30 minutes, twice daily at 8:00 AM and 8:00 PM, for 3 consecutive days at 9 mW/cm2 delivering an energy fluence of 5.4, 10.8, and 16.2 J/cm2 at eye levels, respectively. Subsequently, the 670 nm light was administered once daily at 8:00 AM for 4 consecutive days delivering an energy flu-ence of 2.7, 5.4, and 8.1 J/cm2 at eye levels, respectively. On the other hand, the shame group did not receive any 670 nm light treatment.

  1. I think the element numbers are wrong: e.g. in FVEP the methods have 12/group, but in the figure description only 6/group. On the bar charts, the individual values should also be indicated for each figure.

Ans:

   Thank you for pointing out this important detail.

We clarify that n = 6 rats per group, and FVEP recordings were obtained from both eyes, resulting in n = 12 eyes per group for the FVEP data.

We have revised the schematic diagram to clearly indicate this and avoid any confusion.

In addition, individual values for each subject have already been plotted as data points on the charts.

Are the results presented clearly and in sufficient detail, are the conclusions supported by the results and are they put into context within the existing literature?

  1. Lines 287-289: RGC - sham: 1652.3 ± 210.3/mm2, AION: 587.8 ± 187.3/mm2, RL-10: 911.6 ± 253.5/mm2 in lines 287-289: while the RL-10 group 291 showed a significant increase (1.6-fold) in the RGC density when compared to the sham 292 group - this is a contradiction, on the other hand it did not increase, it was higher, because we are not talking about the same animals.

Ans:  

We thank the reviewer for pointing out this oversight. Lines 372, in the RGC density when compared to the sham group is a typo. It should be compared to the AION group. We have corrected it in line 372 of the revised manuscript.

  1. Figure 3. y axis: relative expression of GAPDH? Not relative to GPDH?

Ans:

    We thank the reviewer for identifying this important detail. To avoid confusion, we have revised the y-axis label to “Relative mRNA expression (Fold change)” in the relevant figure. As correctly noted, GAPDH was used as the internal reference gene in the analysis.

  1. Fig 3 the retinas are wavy, if possible, it would be good to replace with another representative photo.

Ans:

We thank the reviewer for this valuable suggestion. In response, we have replaced the original image shown in Figure 5 with a clearer and more representative photograph showing a properly preserved retinal structure. The updated image better reflects the typical histological morphology and improves visual interpretation.

  1. The representative Western blots do not look nice, replace them with another and send the original blots. How were the results quantified? Was the background extracted? Because it is not uniform, one group has a much larger background than the other.

Ans:

    We thank the reviewer for their valuable comments regarding the Western blot images. As suggested, we have replaced the representative Western blot images with higher-quality images to better reflect the experimental results shown in Figure 3D.

    For densitometric analysis, ImageJ software (NIH, Bethesda, MD) was used. All protein bands were normalized to GAPDH as the loading control. Background signal was subtracted using the "rolling ball" algorithm in ImageJ, and care was taken to apply the same settings across all groups to ensure consistency. Only bands within the linear range of detection were used for quantification. We agree that background variability can affect interpretation, and we have ensured uniform background correction in the updated analyses.

  1. Why is it that after 10 minutes of stimulation, the amplitude measured during FVEP was similar to the control, whereas after 20-30 minutes of stimulation, it was unchanged compared to the rAION group?

Ans:

    We appreciate the reviewer’s thoughtful observation. We have described this in detail in Discussion.

    The lack of neuroprotection observed with 20- or 30-minute 670 nm PBM exposure in the rNAION model may be explained by the concept of “acquired resilience”, as proposed by Stone et al. [77] , offers a compelling explanation for the observed dose-dependent effects of PBM.

The therapeutic effect observed following 10 minutes of 670 nm PBM is likely attributable to optimal activation of endogenous protective pathways, such as mitochondrial enhancement, anti-oxidative responses, and neurotrophic signaling, as part of the acquired resilience mechanism proposed by Stone et al. (Dose Response. 2018 Dec 27;16(4):). This short-duration PBM likely provided a sub-threshold metabolic stimulus that preserved RGCs function and visual conduction, as evidenced by P1–N2 amplitude recovery in FVEP.

In contrast, longer exposures (20–30 minutes) may have exceeded the hormetic window of beneficial stimulation. According to the principle of hormesis, PBM follows a biphasic dose-response curve: low doses stimulate protective mechanisms, while excessive or prolonged exposure may lead to metabolic fatigue, stress pathway desensitization, or even subtle phototoxic effects. As a result, the 20–30 minute treatments failed to further enhance FVEP responses, and the amplitudes remained comparable to those in the rNAION group.

Are all of the cited references relevant to the research?

4.2. RGC Survival: source 42 (2006): it does not include ganglion cells, nor does the clinical trial mentioned, but it does say that it may be good for optic nerve injury I would rather recommend this here: https://pmc.ncbi.nlm.nih.gov/articles/PMC8421781/

Ans:

    We thank the reviewer for the valuable suggestions. We agree with the reviewer’s suggestion and we have replaced the original citation with a more appropriate and updated reference as 54.

Detailed comments

see above at the specific points, minor:

  1. Why is the abbreviation of the model rAION when the abbreviation of the disease is NAION? (At some point it is written as rat NAION model, I think it would be more transparent if it was abbreviated as rNAION.).

Ans:

We thank the reviewer for this helpful suggestion. To maintain consistency and clarity, we have revised the abbreviation throughout the manuscript to “rNAION”, which clearly reflects the rat model of non-arteritic anterior ischemic optic neuropathy (NAION). All instances of “rNAION” have been corrected accordingly in the revised manuscript.

  1. There are some typos: e.g. line 260: in the title function.

Ans:

We thank the reviewer for pointing out this error. The typo in line 340 has been corrected in the revised manuscript.

The manuscript has been revised with attention to English language and readability. Figures were reviewed and adjusted to improve clarity and consistency where necessary.

Reviewer 4 Report

The present paper provides interesting data on the protective effect of 670 nm light exposure in a rAION animal model. The present data confirm and extend previous results obtained in human and animal models of retinal neurodegenerative diseases. Results are clear and well presented, confirming complex ways of actions. Recently, the hypothesis has been advanced that FBM might induced “acquired resilience” (Stone et al 2018) what is the idea of authors about this hypothesis? This might explain why long exposure (20-30 min) is not effective but it may be even dangerous.

Minor points: in VEP recordings latency appears longer in RL-10 group is it real? The first treatment started immediately after the recovery from anesthesia?

Author Response

Major comments

The present paper provides interesting data on the protective effect of 670 nm light exposure in a rAION animal model. The present data confirm and extend previous results obtained in human and animal models of retinal neurodegenerative diseases. Results are clear and well presented, confirming complex ways of actions. Recently, the hypothesis has been advanced that FBM might induced “acquired resilience” (Stone et al 2018) what is the idea of authors about this hypothesis? This might explain why long exposure (20-30 min) is not effective but it may be even dangerous.

Ans:

We thank the reviewer for raising this insightful point. We have described this in detail in Discussion. We agree that the concept of “acquired resilience”, as proposed by Stone et al. (Dose Response. 2018 Dec 27;16(4).), offers a compelling explanation for the observed dose-dependent effects of PBM.

The lack of neuroprotection observed with 20- or 30-minute 670 nm PBM exposure in the rNAION model may be explained by the concept of “acquired resilience”, as proposed by Stone et al. [77] , offers a compelling explanation for the observed dose-dependent effects of PBM. It describes an adaptive system in which low-dose physical or metabolic stressors (such as PBM) trigger endogenous protective responses that increase tissue resistance to subsequent injury. This system is mechanistically dis-tinct from acquired immunity and involves non-specific, organism-wide responses in-cluding mitochondrial enhancement, antioxidant enzyme upregulation, mobilization of bone marrow-derived cells, and trophic cytokine release.

In the context of our study, 670 nm PBM at low energy densities may act as a mild metabolic stressor, activating these defense pathways and thus preserving RGCs func-tion. However, as predicted by the principle of hormesis, this protective response is dose-dependent. Excessive exposure duration (20–30 minutes) may surpass the opti-mal stimulation window, leading to diminished efficacy or potential stress overload. This biphasic response may account for the lack of additional benefit, or even reduced efficacy, observed in longer treatment groups.

 Recent concepts of acquired resilience, provide a plausible explanation for the biphasic dose-response observed in our PBM experiments. Low-dose stressors, such as short-term 670 nm light exposure, may activate mitochondrial and anti-inflammatory pathways to protect retinal neurons, while prolonged exposure may exceed the opti-mal threshold and disrupt homeostasis. This supports the therapeutic principle of hormesis, which is widely recognized in resilience biology.

Detailed comments

Minor points: in VEP recordings latency appears longer in RL-10 group is it real? The first treatment started immediately after the recovery from anesthesia?

Ans:

    Thank you for your careful observation. No, the latency difference was not real. We have replaced the VEP image in the revised version with a more representative and appropriate recording that better reflects the actual group averages shown in Figure 3.

We confirm that the first session of 670 nm PBM was administered 12 hours after recovery from anesthesia. This timing was selected to ensure full physiological stabilization of the animals prior to treatment initiation.

The manuscript has been revised with attention to English language and readability. Figures were reviewed and adjusted to improve clarity and consistency where necessary.

Round 2

Reviewer 2 Report

The authors have addressed all of the reviewers' comments sincerely and thoroughly. I recommend acceptance of the manuscript after minor revisions to correct typographical and grammatical errors. 

Your article would benefit from improvements in grammar, clarity, sentence structure, and tone to ensure it reads smoothly and professionally in a scientific manuscript. Some sentences are repetitive or slightly awkward and need tightening for clarity. Followings are several examples..

line 23 : redundant "Non-arteritic anterior ischemic optic neuropathy (NAION)"

line 104 : repetition "light exposure"

line 109: "As for our experimental schedule"

line 117: "scarified "

line 121 : "Schematic diagram for illustrates"

line 131 : "Each animal had a filter-top cage, and they .."

line 147 : "was intravenously was administered"

line 153 : "for 12 1-s pulses"

etc...

Author Response

Major comments

The authors have addressed all of the reviewers' comments sincerely and thoroughly. I recommend acceptance of the manuscript after minor revisions to correct typographical and grammatical errors. 

Ans:

We sincerely appreciate the reviewer’s positive feedback and recommendation for acceptance. We have carefully revised the manuscript to correct all identified typographical and grammatical errors, as suggested. The revised version has been thoroughly proofread to ensure clarity and accuracy in language.

Thank you once again for your valuable comments and support.

Detailed comments

Your article would benefit from improvements in grammar, clarity, sentence structure, and tone to ensure it reads smoothly and professionally in a scientific manuscript. Some sentences are repetitive or slightly awkward and need tightening for clarity. Followings are several examples..

line 23 : redundant "Non-arteritic anterior ischemic optic neuropathy (NAION)"

line 104 : repetition "light exposure"

line 109: "As for our experimental schedule"

line 117: "scarified "

line 121 : "Schematic diagram for illustrates"

line 131 : "Each animal had a filter-top cage, and they .."

line 147 : "was intravenously was administered"

line 153 : "for 12 1-s pulses"

etc...

Ans:

We thank the reviewer for the valuable comments on grammar, clarity, sentence structure, and tone. In response, we have thoroughly revised the manuscript to enhance readability and scientific professionalism. Beyond the listed examples, we also addressed similar issues throughout the text, refining sentence flow, removing redundancies, and ensuring a clear, consistent tone.

We appreciate the reviewer’s insights, which have helped improve the overall quality of our manuscript.

Reviewer 3 Report

The authors have addressed the questions and suggestions raised, it can now be accepted

The authors have addressed the questions and suggestions raised, it can now be accepted

Author Response

Review 3

Major comments

The authors have addressed the questions and suggestions raised, it can now be accepted

Ans:

We sincerely appreciate the reviewer’s positive feedback and recommendation for acceptance. We are grateful for the opportunity to revise our manuscript and thank the reviewer for their valuable comments, which have contributed to improving the quality and clarity of our work.

Detailed comments

The authors have addressed the questions and suggestions raised, it can now be accepted.

Ans:

We sincerely thank the reviewer for their thoughtful evaluation and kind remarks. We appreciate the acknowledgment that the questions and suggestions have been adequately addressed, and we are pleased that the revised manuscript meets the expectations.

Reviewer 4 Report

the authors responded clearly to all comments raised

no comments

Author Response

Review 4

Major comments

the authors responded clearly to all comments raised.

Ans:

We sincerely thank the reviewer for their positive feedback and for recognizing our efforts in clearly addressing all the comments and suggestions. We are grateful for the constructive review, which has significantly improved the quality of our manuscript.

Detailed comments

no comments.

Ans:

We appreciate the reviewer’s time and consideration. We are pleased that no additional detailed comments were raised. Thank you for your support in the evaluation of our manuscript.